# Compositional boundary layers trigger liquid unmixing in a basaltic crystal mush

Victoria C. Honour [1]*, Marian B. Holness[1], Bernard Charlier [2], Sandra C. Piazolo [3], Olivier Namur[4], Ty J. Prosa [5], Isabelle Martin[5], Rosalind T. Helz [6], John Maclennan[1] & Marlon M. Jean [7,8]

The separation of immiscible liquids has significant implications for magma evolution and the formation of magmatic ore deposits. We combine high-resolution imaging and electron probe microanalysis with the first use of atom probe tomography on tholeiitic basaltic glass from Hawaii, the Snake River Plain, and Iceland, to investigate the onset of unmixing of basaltic liquids into Fe-rich and Si-rich conjugates. We examine the relationships between unmixing and crystal growth, and the evolution of a nanoemulsion in a crystal mush. We identify the previously unrecognised role played by compositional boundary layers in promoting unmixing around growing crystals at melt-crystal interfaces. Our findings have important implications for the formation of immiscible liquid in a crystal mush, the interpretations of compositional zoning in crystals, and the role of liquid immiscibility in controlling magma physical properties.

[1] Department of Earth Sciences, University of Cambridge, Cambridge CB2 3EQ, UK. [2] Department of Geology, University of Liege, 4000 Sart Tilman, Belgium. [3] School of Earth and Environment, University of Leeds, Leeds, UK. [4] Department of Earth and Environmental Sciences, KU Leuven, 3001 Leuven, Belgium. [5] CAMECA Instrument Inc., 5470 Nobel Drive, Madison, WI, USA 53711. [6] United States Geological Survey, MS 926A, Reston, VA 20192, USA. [7] Institut für Mineralogie, Leibniz Universität Hannover, Callinstr. 3, 30167 Hannover, Germany. [8] Department of Geological Sciences, University of Alaska-Anchorage, 3211 Providence Drive, CPSB 101, Anchorage, AK 99508, USA. *email: vch28@cam.ac.uk

The unmixing of mafic magmas into immiscible pairs of Fe- and Si-rich conjugates was first identified in lunar samples[1] and subsequently in terrestrial volcanic and plutonic rocks[2–7]. Given the preferential partitioning of elements of economic interest into the Fe-rich conjugate[8], a detailed understanding of the mechanisms by which significant differences in viscosity, density and wetting properties control the physical behaviour of the unmixed liquids in a crystal mush is essential, to better understand the origin of ore deposits hosted in mafic bodies[9–11]. Such an understanding is also vital to decode the processes responsible for the production of silicic melts by the differentiation of basalts.

Since solidified rocks do not generally preserve a record of parental melts, our ability to track the onset and evolution of magma unmixing has relied on experiments[12–14] and thermodynamic modelling[15]. In this study, we use microscale high-resolution imaging, electron probe microanalysis (EPMA) and nanoscale atom probe tomography (APT) to study a suite of samples from the 1959 Kīlauea Iki lava lake. The samples were drilled from the upper crust of the solidifying lava lake in 1976, 1979, 1981 and 1988, with each core reaching a different depth (Supplementary Note 1; Supplementary Fig. 1). These samples provide a unique opportunity to study the systematic evolution of unmixing during crystallisation in a natural system. The Kīlauea Iki sample suite is complemented by additional examples of glassy tholeiitic basalts from the Laki eruption, Iceland[16,17] and basalt flows from the Snake River Plain (SRP), USA[18,19], which preserve similar microstructural evidence of immiscibility (see Supplementary Figs. 3 and 4; Supplementary Note 1).

## Results

**Principles of unmixing.** The miscibility gap is defined by the binodal curve, which outlines a region on a phase diagram below which it is energetically favourable for a single-phase liquid to unmix. Below the binodal, the spinodal defines the temperature–composition space in which there are no thermodynamic barriers to unmixing and infinitesimally small fluctuations in composition and density grow spontaneously by uphill diffusion[20], leading to phase separation. The interface between the two phases is characterised by low amplitude concentration differences that sharpen with time[21]. The two main mechanisms of unmixing of cooling liquids are therefore nucleation (either homogeneous or heterogeneous) below the binodal of droplets of the volumetrically minor phase or, if the liquid experiences a large, (near-) instantaneous undercooling, by spontaneous spinodal decomposition[20]. Crystal growth also occurs during cooling, with the possibility of the formation of compositional boundary layers (CBLs) in the immediately surrounding liquid, although the effect of this on unmixing has not previously been considered.

**Occurrence of crystal CBLs.** All samples studied contain macrocrysts (>0.5 mm) of clinopyroxene and olivine in a groundmass of plagioclase microcrysts (100–300 μm)[16,17,22] and glass (compositions are presented in Supplementary Table 1). The modal percentage of glass increases linearly with increasing depth in the upper crust of the Kīlauea Iki lava lake; in the 1976 drill core the modal glass percentage increases from 14.3% at 42.8 m depth to 44.4% at 45.5 m[23]. In some samples from all three localities, optimising the contrast-brightness in backscatter electron (BSE) images at high magnification reveals that plagioclase grains are surrounded by a continuous CBL[24], which is enriched relative to the bulk melt with elements incompatible in plagioclase, chiefly Fe, with Mg, P, Mn and Ti. The Fe-rich CBLs are compositionally similar to the Fe-rich immiscible liquid analysed in experimental and natural sample studies[25] (Supplementary Fig. 5).

We examined samples from the 1976 Kīlauea Iki drill core at progressively shallower depths (between 45.5 and 37.6 m) to trace the temporal evolution of the Fe-rich CBLs. Pre-quench temperatures range from 1112 °C to sub-solidus, based on glass thermometry, with absolute calibration uncertainties of ±8 °C[26,27] (Supplementary Table 1; see 'Methods'). Plagioclase crystallisation commenced at 1163 °C[28,29]. As the Kīlauea Iki lava lake cooled, the isotherms moved downwards in the lava lake crust[23,28] and cooling rates decreased with time. Hence, the cooling rates experienced by specific core samples strongly depend on where (and when) they were collected. Samples studied from the 1976 Kīlauea Iki drill core cooled at 0.009–0.011 °C/h, whereas samples from the 1981 and 1979 drill cores cooled at a slower rate (0.0002–0.0003 °C/h)[28].

The Kīlauea Iki plagioclase is typically well facetted (Fig. 1), indicating interface-controlled growth[30]. It is normally zoned (with a maximum compositional difference of ~10 mol. % An[29]), generally exhibiting a core and a well-defined relatively Na-rich rim (identifiable in energy-dispersive X-ray spectroscopy (EDS) and BSE images). The thickness of the Fe-rich CBL varies from 0.5 to 2.5 μm and is thickest on the fastest growing faces of the grain, which are those perpendicular to (010) (cartoon shown in Supplementary Fig. 7; Fig. 1a, b). The aspect ratio of the Fe-rich CBL (defined as the ratio of the thickness of the CBL on the faces perpendicular to (010) and the thickness of the CBL on the (010) faces) positively correlates with the aspect ratio of the plagioclase Na-rich rim (defined as the ratio of the rim thickness on the faces perpendicular to (010) and the rim thickness on the (010) faces; Supplementary Fig. 6, Supplementary Table 2). Both these aspect ratios decrease upwards towards the lava lake surface.

The Fe-rich CBL is most distinct, being both continuous and homogeneous, in samples collected deeper than 44 m (>20% glass) in the Kīlauea 1976 drill core. At 43.7 m (1054 °C), the Fe-rich CBL is discontinuous, comprising sub-micron Fe-rich droplets surrounded by a continuous Si-rich phase (Fig. 1c). By 42.8 m (1017 °C), Fe-rich CBLs are absent and evidence of immiscibility is confined to apparently isolated pockets of glass (generally 1–3 mm² in two-dimensions) in which 5–10 μm diameter Fe-rich droplets are attached to plagioclase with a high apparent wetting angle (>100°; Fig. 1d), together with isolated Fe-rich droplets of comparable diameter dispersed within the Si-rich continuous phase. Locally, at 42.8 m (1017 °C), Fe-rich droplets adhere to pyroxene grains with a low apparent wetting angle (<40°; Fig. 1d).

In samples with thick, >1 μm, Fe-rich CBLs (i.e., the deeper 1976 drill core samples; Supplementary Fig. 8), the contact between the Fe-rich CBL and the interstitial glass appears gradational over 3–6 μm in BSE images, with the margin of the Fe-rich CBL comprising an emulsion of Fe-rich immiscible droplets dispersed in a Si-rich continuous phase: the Fe-rich droplets fine away from the Fe-rich CBL. Shallower drill core samples with thinner Fe-rich CBLs exhibit a sharp contact between the Fe-rich CBL and the surrounding interstitial glass (Fig. 1b). Plagioclase grains in the deepest sample studied (45.5 m) have prominent spines, <25 μm long, extending in the direction of fastest growth, and these indicate a period of diffusion-limited growth[31] (Fig. 1a, b). The spines are rarer and shorter (<1 μm long) in shallower samples with thinner Fe-rich CBLs. Pyroxene is surrounded by a 1–2 μm-wide Si-rich CBL (Si-rich CBL) which, although less well developed than the Fe-rich CBL around plagioclase (Fig. 1c), mirrors the attributes of the latter.

While each sample experienced the same rate of quenching, the character of the Fe-rich CBL systematically evolves through the

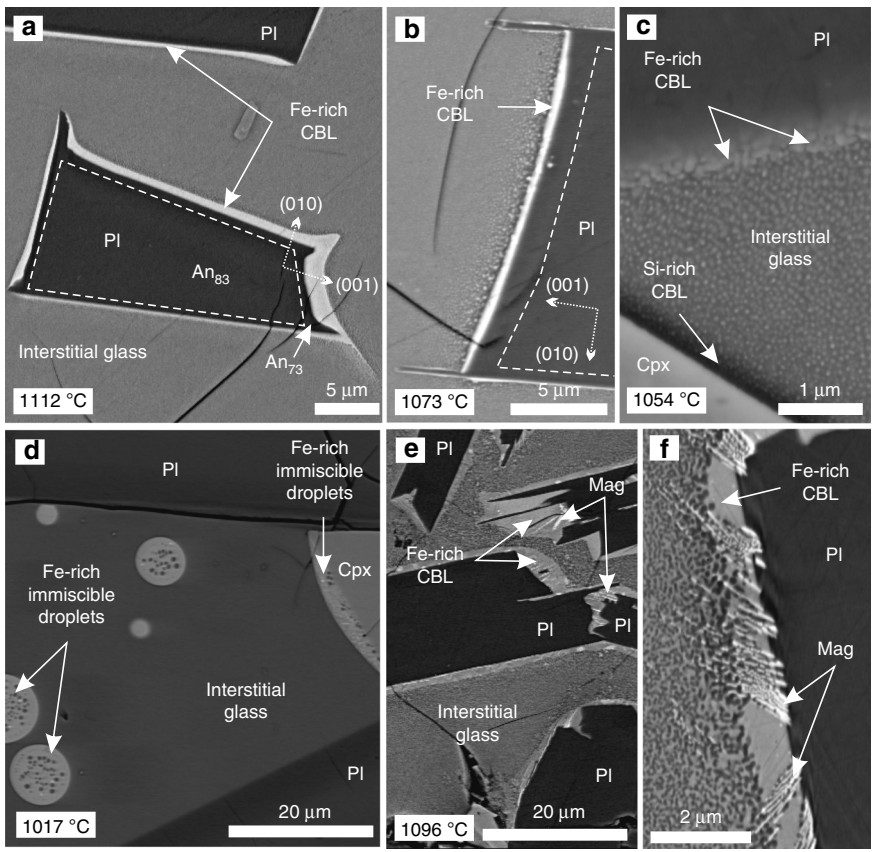

**Fig. 1** Compositional boundary layer morphology in BSE images from samples of the 1976 Kīlauea Iki lava lake. **a–d** The changing morphology of immiscible liquids in the 1976 drill core from the Kīlauea Iki lava lake, viewed in BSE images. **a** Plagioclase lath (Pl) surrounded by an Fe-rich compositional boundary layer sampled at 45.4 m depth in the crust (quench temperature of 1112 °C). **b** A thinner compositional boundary layer surrounding plagioclase (quench temperature of 1073 °C) and the nanoemulsion is coarser. **c** At 43.7 m depth (quench temperature of 1054 °C), the compositional boundary layer surrounding plagioclase is discontinuous and sub-micrometre thick. Note the presence of darker (more Si-rich) glass immediately surrounding pyroxene (Cpx) grains. The bulk of the glass comprises a nanoemulsion. **d** Isolated and attached Fe-rich droplets in interstitial glass at 42.8 m depth (quench temperature 1017 °C). Note the Fe-rich liquid wets the pyroxene with a low wetting angle. **e**, **f** Compositional boundary layers surrounding plagioclase from Snake River Plain tholeiites. The calculated natural quench temperature of this sample is 1096 °C. Note the oxide dendrites within the compositional boundary layer and partially embedded in the plagioclase

1976 and 1981 Kīlauea Iki drill cores. Hence, the formation of the CBL cannot be related to drilling-induced quenching. This interpretation is supported by a number of observations. First, plagioclase grains are surrounded by an Fe-rich CBL regardless of their position relative to the drill core edge (and therefore the rate of quenching). Second, the Na-rich plagioclase rim width correlates with Fe-rich CBL thickness (Supplementary Fig. 6; Supplementary Table 2). Third, the Na-rich rim aspect ratio is consistent with slower growth than that expected during a quench[32]. Fourth, locally, the Fe-rich CBL contains dendritic crystals of oxide partially embedded in the plagioclase substrate, consistent with simultaneous oxide and plagioclase growth (Fig. 1e, f).

Similar Fe-rich CBL microstructural features are observed in the glass of the SRP tholeiites from the Sugar City drill core and in the Laki samples preserved by a natural quench as the lava flows cooled rapidly through the glass transition temperature (Supplementary Figs. 2 and 3). This occurred at temperatures between 1093 and 1096 °C for the SRP basalts and at 1114 °C for the Laki sample (based on glass thermometry[26]—see 'Methods') comparable to the drill-related quenching of the 1981 Kīlauea Iki drill core samples at 1114–1140 °C[27]. In the 1981 Kīlauea Iki drill core, the Fe-rich CBL is thin yet well defined, thinning with decreasing drill core depth over 8.2 m, from an average of 0.8–0.5 μm for the Fe-rich rim perpendicular to the (001) plagioclase face

(Supplementary Fig. 4). The SRP Fe-rich CBL is well defined, thinning with decreasing drill core depth over 5 m, from 2.0 to 1.5 μm for the Fe-rich rim perpendicular to the (001) plagioclase face (Supplementary Fig. 2), while the Laki sample has more diffuse, thinner Fe-rich CBLs (~0.7 μm) (Supplementary Fig. 3).

**Kīlauea Iki glass compositions.** With decreasing depth in the 1976 drill core relative to the lake surface (from 45.5 to 42.8 m depth; with corresponding quench temperatures of 1112 and 1017 °C, respectively), $SiO_2$ concentrations of the apparently homogeneous bulk liquid (see 'Nanoemulsion formation') increase from 50 to 65 wt.%, while $FeO_{total}$ concentrations decrease from 12 to 5 wt.% (Supplementary Fig. 9, Supplementary Table 1; see 'Methods'). Correspondingly, the FeO concentration of the Fe-rich droplets increases with decreasing depth: at 45.5 m (1112 °C) the average FeO concentration is 21 wt.%, increasing to 27 wt.% at 42.8 m depth (1017 °C), consistent with published compositions for immiscible liquids[12] (Supplementary Fig. 5). There is no change in the crystal assemblage associated with these variations in glass composition. Furthermore, because the Kīlauea Iki lava lake system was open with respect to $H_2O$ and had a confining pressure typically <6 bars[28], the observed compositional variations cannot have resulted from variations in either $H_2O$ content or pressure.

EPMA compositional profiles extending 50 µm outwards from the Fe-rich CBL show constant $SiO_2$ concentrations, whereas oxides such as FeO, $TiO_2$ and MgO are depleted in a 10 µm wide zone adjacent to the Fe-rich CBL (Fig. 2; Supplementary Table 3); these are oxides that preferentially partition into an Fe-rich immiscible liquid[13]. In the same 10 µm wide zone, there is a small increase in elements such as $Al_2O_3$ and $Na_2O$; these are oxides that preferentially partition into a Si-rich immiscible liquid (Fig. 2). This chemical concentration pattern is expected for the formation of a two-phase mixture by nucleation and growth[33].

**Atomic-scale chemical and spatial characteristics.** Three-dimensional APT reconstructions of compositional data (Supplementary Fig. 10; Supplementary Movie 1) of the Kīlauea Iki lava lake 1976 drill core sample from 44.8 m (1090 °C[27]) show that the

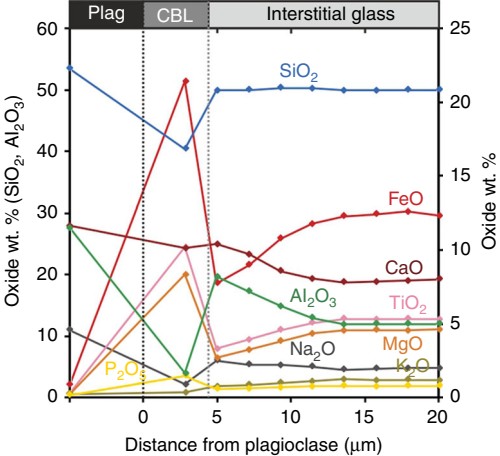

**Fig. 2** Micron-scale compositional changes across an Fe-rich compositional boundary layer (CBL). A micron-scale compositional transect obtained using EMPA through Si-rich glass adjacent to an Fe-rich CBL. Sample from the crust of the Kīlauea Iki lava lake (quench temperature of 1112 °C), sampled by the 1976 drill core at 45.5 m. Note the two different scales depending on element

Fe-rich CBL surrounding plagioclase grains is internally homogeneous (Supplementary Fig. 11), with a sharp contact against plagioclase (Supplementary Fig. 12) and a diffuse contact over 10 nm with the adjacent liquid (Fig. 3; Supplementary Table 4). There is no marginal Fe-enrichment of the plagioclase, even at the nanoscale (Fig. 3). The Fe-rich CBL is rich in those elements expected to preferentially partition into an Fe-rich immiscible liquid[13]. The APT compositional data show distinct jumps between the different components with no indication of diffusion-related elemental profiles (Fig. 3), therefore we discount significant post-solidification diffusion of elements between the two glasses.

EPMA traverses across the interstitial liquid surrounding the Fe-rich CBLs (from the same sample) show a homogeneous composition (Supplementary Table 3). At the greater spatial resolution provided by the APT, it is evident that the interstitial liquid (up to 14 µm away from the Fe-rich CBL; Supplementary Fig. 10) is actually a nanoemulsion in which the Fe-rich immiscible liquid forms branch-like interconnected structures and isolated clusters within the continuous interstitial liquid. The Fe-rich immiscible liquid is separated from the interstitial liquid by sharp boundaries (glass tip 179377, shown in Fig. 3), with no quantifiable spatial variations in morphology, composition, size or spacing. Concentrations of Si increase within 2–4 nm of the nanoemulsion phase boundary on the Si-rich side; likewise, Fe concentrations increase on the Fe-rich side (Fig. 3). Importantly, these compositional features are present regardless of the size of the individual regions; they are consistent with spinodal decomposition[33].

The Fe-rich CBL and the Fe-rich immiscible liquid of the nanoemulsion are largely similar in composition, but the Fe-rich CBL is depleted in Al and Na relative to the Fe-rich immiscible liquid of the nanoemulsion: the Al and Na concentration of the Fe-rich immiscible liquid of the nanoemulsion lies on a mixing line between plagioclase and the Fe-rich CBL (Supplementary Fig. 13; Supplementary Tables 1 and 4).

**Nucleation and evolution of unmixing: triggers and relationships.** The non-zero wetting angle[34] of Fe-rich liquid on a non-mafic mineral and the spatial correlation between Fe-rich CBL thickness and the growth rates of different plagioclase faces (Fig. 1, Supplementary Fig. 6) both demonstrate that the Fe-rich

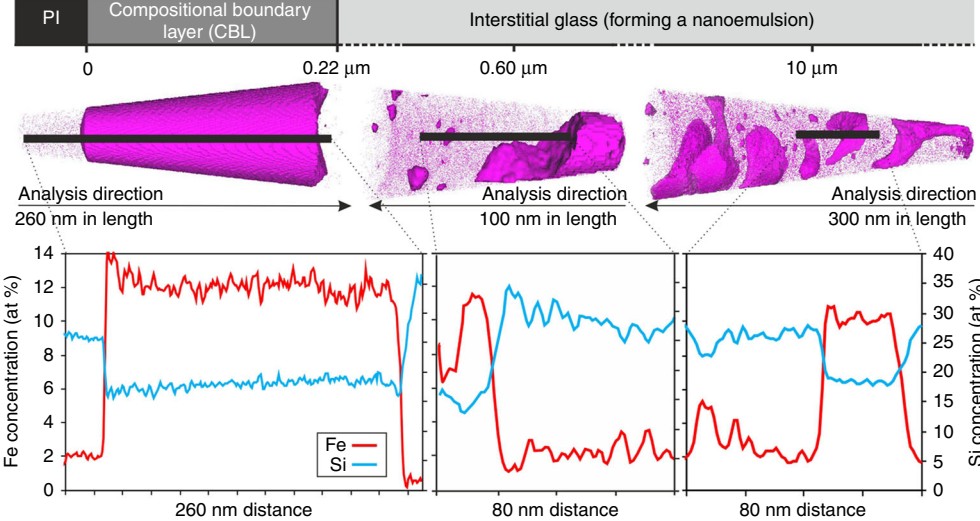

**Fig. 3** Three-dimensional reconstruction of atom probe tomography (APT) data from compositional boundary layers around plagioclase. APT reconstructed data from three representative APT tips including 1D element profiles across compositional boundaries; Kīlauea Iki lava lake sample from 44.8 m depth (quench temperature of 1090 °C). Each dot represents a single atom, but not all the atoms are shown. One-dimensional compositions from sub-volumes represented by black bars are plotted for Fe and Si concentrations (at. %) for each APT tip. A movie of an ATP tip is shown in Supplementary Movie 1

CBL surrounding plagioclase, and the weakly developed Si-rich CBL surrounding pyroxene, are disequilibrium features created during crystal growth. That the Fe-rich CBLs are continuous, with no evidence of separation into individual droplets required for textural equilibrium, must therefore be due to continuing plagioclase growth sustaining the Fe-rich CBL and preventing it from reaching textural equilibrium.

The creation and maintenance of a CBL around a growing crystal requires plagioclase growth rates to be commensurate with, or surpass, the diffusion rates of elements in the surrounding liquid. Different elements have different diffusion rates in different viscosity melts[35]; however, diffusion coefficients for the Makaopuhi and Alae lava lakes[36] range from $5 \times 10^{-15}$ to $5 \times 10^{-14}$ m²/s—these are lava lakes of similar composition to the Kīlauea Iki lava lake but smaller in size. The Makaopuhi and Alae lava lakes have plagioclase growth rates of ~$5 \times 10^{-11}$ to $10^{-12}$ m/s[36]; we calculate slightly slower plagioclase growth rates of c. $10^{-12}$ m/s, which is unsurprising given the samples are from a bigger lava lake and are sampled from deeper in the lake (see 'Methods'; Supplementary Table 2). Over a distance of 20 μm, the plagioclase growth rate is comparable to that of the diffusion rate.

Previous studies suggest that cryptic crystal chemical zonation in plagioclase on the 1 μm scale may result from CBL development[24]. Our data showing the Na-rich rims on the same scale as the surrounding Fe-rich CBLs (Fig. 1a; Supplementary Table 2), and the relatively Na- and Al-poor composition of the Fe-rich CBL compared with the nanoemulsion Fe-rich liquids (Supplementary Table 4), are consistent with the plagioclase Na-rich rim forming at the same time as the Fe-rich CBL. This may lead to a cyclic process of formation and destruction[24], whereby the development of Fe-rich CBLs in a continuously evolving system facilitates the growth of Ca-poor plagioclase, which then reverts to relatively Ca-rich growth if the CBL is destroyed by magma flow. The morphology of the plagioclase grains from the Kīlauea Iki lava lake (Fig. 1; along with the SRP data set, Supplementary Fig. 4) suggests that once the Fe-rich CBL has formed, crystal growth switches from interface-controlled to diffusion-limited, resulting in the growth of extended spines on the grain corners where the Fe-rich CBLs are less depleted in crystal-forming elements[24,37].

The Fe-rich CBL surrounding plagioclase becomes thinner as solidification proceeds (Supplementary Fig. 8; Supplementary Table 2), and the Fe-rich CBL evolves during solidification into a series of attached Fe-rich immiscible droplets (at 42.8 m, 1017 °C, Fig. 1d) indicative of an approach to textural equilibrium. This suggests the rate of plagioclase growth, and hence the extent to which textural equilibration is prevented, decreases with progressive crystallisation and liquid evolution, perhaps due to a change in cooling rate. Isolated Fe-rich droplets in the shallowest (and most solidified) Kīlauea Iki lava lake sample provide further evidence of a temporal evolution in the unmixing process (the idealised evolution is illustrated in Fig. 4). This is consistent with unmixing later in the crystallisation history occurring by homogeneous nucleation in the bulk liquid, rather than being triggered by crystal growth, plausibly because only later in the crystallisation history is the undercooling sufficient to drive homogeneous nucleation.

The 1976 Kīlauea Iki lava lake transect covered by our sample suite cooled at 0.009–0.0011 °C/h[28]. If we assume a constant cooling rate, the temporal difference between the deepest (45.5 m) and the shallowest (42.8 m) samples containing immiscible microstructures is of the order of 11–14 months, suggesting that the differences we observe in the distribution and morphology of the unmixed Fe-rich liquids developed over a year. As the isotherms moved downwards with time and the slope of the

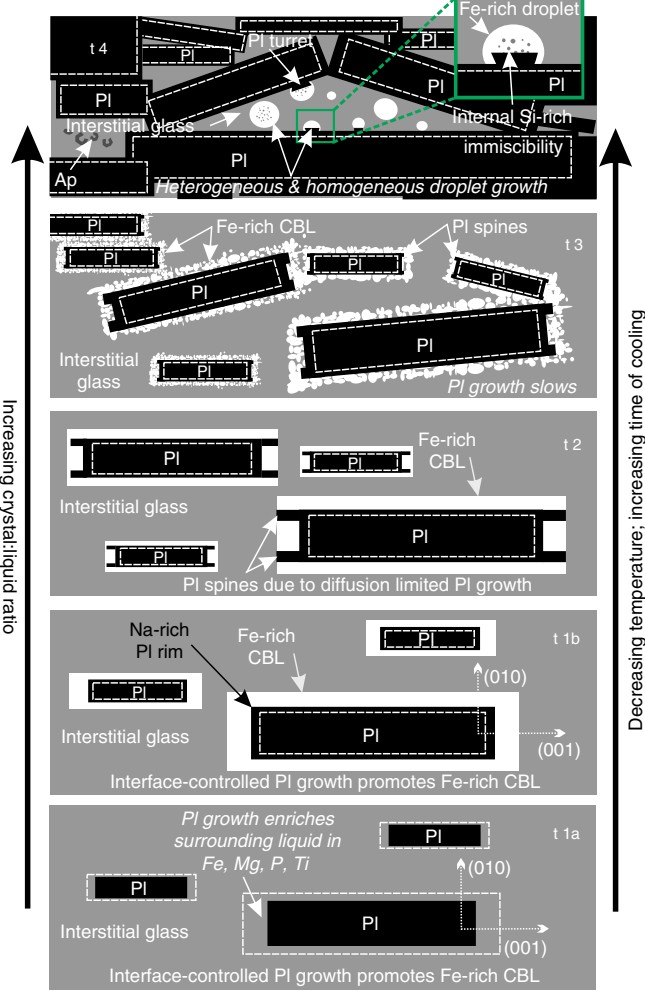

**Fig. 4** Schematic evolution of unmixing features with progressive cooling at natural rates. Interface-controlled plagioclase growth promotes the formation of an Fe-rich compositional boundary layer, due to enrichment of the surrounding liquid in metalliferous elements. The Fe-rich CBL evolves temporally, coeval with the increasing crystal:liquid ratio, as a function of time and temperature, from panel t1 to t4, where t4 represents the lowest temperature and longest time of cooling. Overall, the creation of an Fe-rich CBL enriched in components rejected by plagioclase promotes full unmixing in liquids close to the binodal, without the need to nucleate and grow droplets. As plagioclase growth slows, an Fe-rich CBL can no longer be maintained and Fe-rich droplets nucleate both homogeneously and heterogeneously. Note that the processes illustrated in the cartoon are italicised

thermal gradient of the lava lake became shallower[28], this estimate is likely to be a maximum.

Although compositional heterogeneities in liquid surrounding growing crystals are typically associated with diffusion-limited crystal growth[31], our observations show that they may also result from growth under interface-controlled conditions (shown by the equant plagioclase morphology)[24] (Fig. 4; t1a and t1b). Furthermore, the creation of an Fe-rich CBL enriched in components rejected by plagioclase promotes full unmixing in liquids close to the binodal, without the need to nucleate and grow droplets. The same is true for the Si-rich CBL developed around pyroxene (Fig. 1c). Our observations thus demonstrate that kinetic effects related to crystal growth strongly affect the location and nucleation of unmixing, particularly at temperatures >1020 °C.

**Nanoemulsion formation**. The observed nanoemulsion in the Kīlauea Iki lava lake drill cores may have formed as the water used during drilling encountered each sample, undercooling sufficiently that the spinode was attained before homogeneous nucleation occurred. Alternatively, it could be a preserved feature of unmixing during the solidification of the Kīlauea Iki magma at natural rates, which has been hitherto missed as these nanoemulsions are only visible with the spatial resolution of APT. The natural cooling rates of the SRP and Laki samples (Supplementary Note 1) may have been sufficient to promote spinodal unmixing, and future work will need to focus on such fine-scale unmixing in both natural and experimental samples, complementing work on nanoemulsions formed by quenching from high temperatures during meteorite impacts[38].

**Effect of crystal growth on the position of the binodal**. An outstanding question is why Fe-rich CBLs, which are such a prominent feature of the samples from Laki, SRP, and Kīlauea Iki (Fig. 1; Supplementary Figs. 2–4), have not been commonly observed in experimental studies of immiscibility in basaltic glass. Depending on the contrast-brightness optimisation for BSE imaging, such Fe-rich CBLs can easily be overlooked. In addition, we suggest that the generally high nucleation density in experimental charges results in a high crystal surface:liquid ratio, which prevents the development of steep compositional gradients in the surrounding liquid. We propose that, at slow cooling rates typical of natural basaltic magmas, a lower nucleation rate, larger crystals, and thus a low crystal surface:liquid ratio, mean that kinetic effects associated with crystal growth lead to earlier (higher temperature) unmixing of an Fe-rich immiscible liquid (in the form of an Fe-rich CBL), compared to experimental analogues. The spread in our compositional data for the Fe-rich CBL and Fe-rich immiscible droplets (Fig. 5; Supplementary Fig. 14) is attributed to limits on the spatial resolution of the EPMA (see Methods).

Our calculated liquid temperature estimates (see 'Methods') were used to constrain the position of the binodal, plotted with respect to the parameter NBO/T (Fig. 5; where NBO is the number of non-bridging oxygens and T is the concentration of the tetrahedrally coordinated network-forming cations Si, Al, P and Ti[39]). The highest temperatures at which Fe-rich CBLs are present in samples from Laki, SRP and Kīlauea Iki are 1114, 1096 and 1112 °C, respectively; this is significantly hotter than previous suggestions for the top of the binodal at 1020 °C[12], although closer to the temperature of ~1060 °C suggested for the onset of immiscibility in the Upper Zone of the Bushveld Complex in South Africa[40]. Our calculated temperature at which homogeneously nucleated Fe-rich immiscible droplets appear (around 1020 °C; Fig. 5) broadly correlates with the temperatures experimentally derived for the onset of immiscibility[12], supporting the hypothesis that the discrepancy between our observations and those of experimental studies is indeed caused by crystal growth in natural samples promoting Fe-rich CBL formation, and hence unmixing, at temperatures higher than required for homogeneous nucleation of droplets.

The timing and extent of Fe-rich CBL development in natural samples depends on magma composition and the kinetics of diffusion and crystal growth. Our work suggests that, if pre-existing phenocrysts act as nuclei, interface-controlled growth at high temperatures could result in immiscibility occurring at a higher temperature than previously recognised.

**High-temperature unmixing: its importance for phase separation and chemical evolution**. Our study provides direct evidence from natural samples of the morphological evolution of

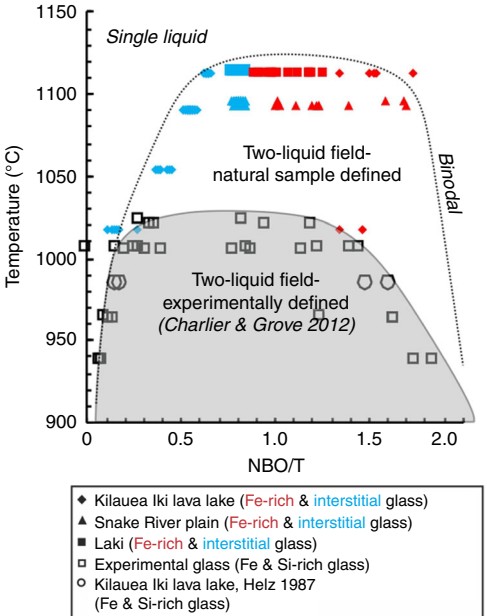

**Fig. 5** The silicate liquid immiscibility field. Temperature as a function of the degree of polymerisation (NBO/T: NBO non-bridging oxygens and T = Si + Al + P + Ti), calculated from EPMA data on the glasses. The shaded grey region represents the experimentally defined two-liquid field[12], the dotted line represents the maximum compositional space where the natural examples show evidence of unmixing by a compositional boundary layer

an unmixing basaltic liquid with progressive cooling (Fig. 4). Our data demonstrate the importance of crystal growth-induced spatial heterogeneities in liquid composition in triggering unmixing. Our application of APT on natural glass gives a unique insight into the three-dimensional (3D) nanoscale morphology of two unmixed liquids at a high spatial resolution (Fig. 3). In our slowly cooled natural samples, the development of an Fe-rich CBL illustrates unmixing at a considerably higher temperature (and lower corresponding undercooling) than observed in experimental charges (Fig. 1), with a progressive attainment of the expected equilibrium morphology of the emulsion. It is clearly not safe to assume that the onset of unmixing is controlled by thermodynamics alone: kinetic factors play an important role.

The calculated density difference between the Fe-rich immiscible liquid (both Fe-rich CBL and Fe-rich immiscible droplets) and Si-rich conjugates examined here[41] is small (<0.3 g/m³), as they are in close proximity to the apex of the miscibility gap[42] (Fig. 5; Supplementary Table 1). This similarity means that separation of the two conjugates is likely to be inefficient. However, as an immiscible Fe-rich end member evolves down-temperature from the apex of the binodal, the density difference will increase. Our study provides direct evidence of a continuous film (in textural disequilibrium) of Fe-rich liquid-coating plagioclase grains (Fig. 1). Conceivably, when the density of the Fe-rich liquid reaches a critical density threshold, the continuity of this film could facilitate the downward movement of the dense, inviscid liquid. Consequently, the onset of unmixing at higher temperatures than previously thought increases the potential for the relative movement of the two immiscible liquids in a gabbroic crystal mush, as at higher temperatures there are lower crystal fractions, and hence mush permeability is higher.

Finally, we note that Iceland and the SRP are tholeiitic provinces with a dearth of intermediate compositions[4]. It is thought-provoking to consider whether the identification of a mechanism for higher temperature unmixing at lower crystallinities has relevance to the formation of the Daly gap.

## Methods

**Scanning electron microscopy (SEM).** Samples were characterised using microscopy and a Quanta FEG 650F SEM for semi-quantitative mineral compositions, element EDS X-ray maps and BSE imagery, set to 10 kV at spot 3, with a working distance of 10 mm at the Department of Earth Sciences, University of Cambridge. The BSE imaging of the interstitial liquid (to identify whether it is comprised of a nanoemulsion) is at the spatial resolution limits of the SEM. Plagioclase aspect ratio, outer growth rim thickness and Fe-rich CBL thickness were measured using BSE images.

**Electron probe microanalysis.** Glass and plagioclase mineral compositions were measured by EPMA using a Cameca SXFiveFE with five WDS spectrometers at the Department of Geology, Mineralogy and Geophysics, Ruhr-Universität Bochum and a Cameca SX-100 with five WDS spectrometers at the Department of Earth Sciences, University of Cambridge, both using the PeakSight software with ZAF correction. The glass analytical routine was run at 15 kV and 8 nA with a 2 μm beam; the plagioclase analytical routine was run at 15 kV and 15 nA with a 2 μm beam. Appropriate glass and silicate mineral secondary standards were analysed. Analyses of Fe-rich immiscible droplets and Fe-rich CBLs were challenging due to the small area presented (<25 μm²) and the unknown 3D shape creating uncertainty about the electron beam interaction volume (this was modelled using the *Casino v2.48* software[43]; Supplementary Fig. 15). This means that the composition of the two immiscible conjugates do not perfectly define the theoretical miscibility gap; the Fe-rich conjugate analyses have a component of mixing with the Si-rich conjugate due to the minimum volume with which the probe beam interacts, therefore we take the end-member compositions. In the Kīlauea Iki lava lake sample from the 1976 drill core at 45.5 m depth below the lake surface, three continuous EPMA line profiles were taken across the glass, extending 50 μm from the Fe-rich CBL surrounding the plagioclase (Supplementary Table 3). In the sample from 44.8 m depth below the Kīlauea Iki lava lake surface, five continuous EPMA profiles were collected outwards from the Fe-rich CBL for comparison with the constant bulk APT composition across the 14 APT tips analysed from the interstitial glass (Supplementary Fig. 10). The EPMA compositional profiles are flat, consistent with the APT compositional analyses.

*Glass thermometry.* For consistency, liquid temperature estimates for the samples from Laki, SRP and Kīlauea Iki were recalculated using MgO glass compositions from EPMA and the relationship $T$ (°C) = 26.3MgO$^{liq}$ + 994.4 °C[26], and where possible, checked against published values[27,44]. This equation was developed from previous geothermometry work based on samples drilled from the Kīlauea Iki lava lake. All temperatures referred to are the pre-quench temperature of the liquid.

**Plagioclase growth rates.** The rate of plagioclase growth in directions parallel to (010) is calculated from the slope of plots of the largest long axis plagioclase crystal in each sample versus time[36]. Time is determined by assuming a constant cooling rate for samples from the KI76 drill core (of 0.009–0.0011 °C/h[28]) from the onset of plagioclase nucleation at 1163 °C[29] to the calculated temperature of the sample. The growth rate is an estimate, as we assume that there is a progressive cooling of the Kīlauea Iki lava lake, where the downward movement of isotherms progressively changed the slope of the geotherm[23,28], resulting in a non-linear cooling rate. The growth rate of plagioclase is anisotropic; the crystals in the lava lake are euhedral, growing in a melt-rich environment, so rapid growth in directions parallel to (010) resulted in limited impingement on adjacent grains (Supplementary Fig. 7). To account for the crystal growth in two directions, the growth rate is halved[36]. There are many sources of error, including those associated with measuring the plagioclase long axis, determining the largest crystal and those converting temperature differences to time scales.

**Sample preparation for APT.** A dual-beam focussed-ion-beam (FIB)/scanning-electron-microscope (FEI Helios 660 with EasyLift™ micromanipulator) was used to fabricate APT specimens. A platinum-based gas injection system was used for specimen protection and attachment. Standard lift-out methodology was used to transfer material wedges from the sample to a micro-tip-array carrier-coupon[45]. Specimens were made sharp using standard annular milling methods[46] with an additional low-energy milling step (5 kV clean-up)[47]. This process was sufficient to remove any FIB-deposited Pt that was used during the lift-out process.

**APT measurements and analysis.** Twenty APT tips were analysed at CAMECA Instruments Inc., Madison, Wisconsin, USA, from samples prepared from 44.8 m depth in the 1976 drill core from the Kīlauea Iki lava lake (Fig. 1e; 1090 °C[27]). Fourteen samples examined the interstitial glass on a 14 μm long traverse away from a plagioclase grain (Supplementary Fig. 10). In addition, six APT samples analysed the plagioclase and the Fe-rich CBL (Supplementary Fig. 10).

The measurements were performed on a LEAP® 5000 XR in laser-pulse mode. Typical analysis conditions were: specimen base temperature of 30 K, constant detection rates between 5 and 10 ions per 1000 pulses, laser pulse energy of 200 pJ, and a pulse frequency set to achieve 2000 Dalton, time-of-flight range (typically 125–250 kHz); a typical voltage curve is shown in Supplementary Fig. 16. Standard IVAS$^{TM}$ shank reconstruction was used for all the reported data[48]. Prepared

specimens typically resulted in needles with consistent taper angles (~10 degrees). For one of the analyses completed before specimen failure, the SEM-measured analysis depth was used to calibrate the APT tip reconstruction. Using default k-factor and image compression factor values, the 10 degree shank angle required ~26 V/nm to provide an initial reconstruction radius and reconstructed reconstruction depth that was consistent with SEM estimates. These reconstruction parameters were subsequently used for all reconstructions.

APT mass ranging for a representative sample is detailed in Supplementary Fig. 17. It is important to note that APT is most suitable for *relative* compositional data rather than absolute compositional data due to issues with mass ranging, efficiency of recovery and complexation during analysis[49].

*IVAS$^{TM}$* 3.8 was used to extract linear one-dimensional (1D) concentration profiles from within sub-volumes of the 3D data sets (plotted in Fig. 3). Profiles were taken across the Fe- and Si-rich immiscible conjugate boundaries of the nanoemulsion parallel with the analysis direction (z-axis) of the glass tip, to ensure maximum spatial resolution and prevent artificial depletion. To delineate these immiscibility boundaries, a 6 at. % Fe atomic concentration isosurface was constructed to define the Fe-rich phase of the nanoemulsion, i.e. the shape/morphology of the Fe-rich immiscible conjugate of the nanoemulsion. The data for these linear 1D compositional profiles stem from an elongate bar with a square profile of 10 nm by 10 nm and the length as shown on the profiles' x-axis (see Fig. 3). This enabled us to choose a 1D profile that was perpendicular to the interface of the Fe- and Si-rich immiscible conjugate boundaries and minimise any smearing artefacts in the compositional profile. These chemical profiles showed compositional depletion and enrichment characteristics typical of spinodal decomposition[33].

## Data availability

The data supporting the findings of this study, as shown in Figs. 1, 2, 4, and 5 and Supplementary Figs. 1–15 are provided as Supplementary Data Tables 1–4 within the paper. The raw APT data that support the findings shown in Fig. 3 and Supplementary Figs. 16 and 17 are available from the corresponding author upon reasonable request.

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

## Acknowledgements

V.C.H. is funded by an NERC DTP studentship project (Grant Ref: NE/L002507/1). B.C. is a Research Associate of the Belgian Fund for Scientific Research-FNRS. We are grateful to Iris Buisman and Giulio Lampronti at the University of Cambridge and Niels Jöns at the Ruhr-Universität Bochum for SEM and EPMA help and advice. M.M.J. thanks Linda Davis and Mary Hodges at Idaho National Laboratory for sampling of the Sugar City drill core and additional support was provided by the Alexander von Humboldt Research Fellowship for Postdoctoral Researchers to M.M.J. This manuscript was significantly improved following thorough and constructive reviews by Steve Parman, Malcolm J Rutherford and an anonymous reviewer.

## Author contributions

V.C.H., M.B.H., B.C. and O.N. initiated the project. R.T.H., J.M. and M.M.J. provided samples, sample descriptions and discussions about the paper. V.C.H. carried out the image analyses of the samples, SEM and EPMA work, APT data analysis and wrote the majority of the paper. O.N. carried out EPMA analyses of the samples and assisted in discussions with B.C. interpreting the data. T.J.P. and I.M. performed data analysis and data reduction of APT data. S.C.P. assisted with APT data analysis, interpretation and extensive discussions revising the manuscript. M.B.H. supervised the project and edited the manuscript. All authors reviewed and approved this paper.

## Competing interests

The authors declare no competing interests.
