## [Peer Review File · Nature Communications]

Reviewers' comments:

Reviewer #1 (Remarks to the Author):

General comments – This paper presents new micro and nano analyses relating to liquid immiscibility in crystal-rich basaltic rocks from three localities: Hawaii, Iceland and the Snake River Plain. The authors find that plagioclase crystals in some samples are surrounded by an Fe-rich compositional boundary layer (CBL) up to a few microns thick. SEM and atom probe imaging of the glass outside of these CBLs is a nanoemulsion of Fe-rich droplets in a Si-rich melt. The conclusion is that the CBL formed due to fast crystallization of the plagioclase and that the ensuing Fe-enrichment produced liquid immiscibility that formed the nanoemulsion. The analyses include the first application of atom probe tomography (a relatively new analytical method) to immiscibility in basalts. The paper is convincing that liquid immiscibility occurred in these samples by the mechanism proposed (crystal growth and spinodal decomposition).

How widespread this process is, and how it relates to fractionation of basalts and the 'Daly gap' is another question. Such extreme CBLs appear to be pretty rare. The CBLs in the samples are up to a few microns width and so would be observable with a standard microprobe (Figure 2a) if they were present in a sample. I think people would have seen them in more localities if they were common. I've not seen them. So not sure that the presence of CBLs in these three samples convinces me this is a widespread process. Also, the CBLs are very thin, don't show a lot of evidence for migrating and appear to only form when the samples are highly crystalline. So how they would ever migrate to form large enough magma bodies to explain the Daly gap is also not clear.

So, I think the study shows an example of interesting thermodynamic (spinodal decomposition) and kinetic (CBL) effects, but I am unconvinced of their connection to larger magmatic processes. Maybe the CBLs or emulsions can be used to extract cooling rates in magmas where they are found?

Specific comments -

Terminology - The terminology should be simplified and used more consistently in the text and figures. For example, in the text, the liquid layer around the crystals is called a CBL (compositional boundary layer), but in Figure 1, it is labeled 'Fe-rich boundary layer', which is not the same as the 'Fe-rich conjugate' found in the emulsion. Similarly, the glass around the crystals (and CBL) is referred to as 'interstitial liquid' in some places in the text (line 124) but labeled as 'Si-rich conjugate' in some panels, and 'nanoemulsion' in other panels. And that is not to be confused with the 'Si-rich boundary layer', which only occurs around the pyroxenes. I suggest trying to simplify the terminology. For the compositional layers around the crystals, maybe try to consistently use the abbreviation CBL, as in text. Could use 'Fe-rich CBL' for the plagioclases and 'Si-rich CBL' for the pyroxenes in the figures. I would drop the use of 'Si-rich conjugate' as it suggests it is different than the interstitial liquid. I suggest using 'interstitial liquid' to refer to the glass outside the CBL, including the labeling in the images. For example, in Figure 2, 'Si conjugate' would be replaced by 'interstitial liquid'. And you can just refer to the compositional gradients in the interstitial liquid that were produced by the formation of the CBL. It is not clear that the formation of the CBL or the nanoemulsion significantly increased the Si in the interstitial liquid. I don't see it in figures 2 or 3. For the Fe-rich immiscible liquid droplets, could use Fe-rich conjugate. But might be clearer to a wider audience if you call them immiscible liquids, which is what most people think of. So something like 'Fe-rich immiscible droplets'.

APT analysis – APT analysis is pretty new for geoscience. So exactly how to present the data, which included millions of data, is still being worked out. At some level, ideally would want the RHIT and range files, and the shank angle for the reconstruction. That's what you would need to reproduce, for example, Figure 3. Given that most people don't have IVAS, maybe that can be an

'upon request' deal. For the interested researcher, I think giving the voltage curve for the analysis would be good. It tells you a lot about how well things went during evaporation. And the mass/charge spectra. For example, the interstitial glass seems to have about 12% FeO (Figure 2), but the Fe concentration in the left panel of Figure 3 drops to almost zero in the interstitial glass. What mass/charge range was used for this profile? Fe+1? Fe+2? FeO+1? Fe₂O₃? Would be useful to know. So I'd suggest including those, as well as the length of the needles, the tip radius and shank angle. This will help future researchers wanting to do similar analyses decide how to make their tips.

Ideally, would use proxigrams to get the compositional profiles across the boundaries, rather than 1D profiles. The difference is that a proxigram can be used to calculate the compositional gradient perpendicular to the surface at all points. This should give the most accurate profile. The 1D profile does not do this, and just averages flat slices. If the boundary is curved, or not orthogonal to the direction of the 1D profile, this will artificially smear out the boundary. I think you can see this effect in the right panel of figure 3. The boundary between the Fe and Si-rich conjugates is thinner on the left than on the right, because the right boundary is more tilted with respect to the 1D profile direction. If proxigrams aren't used, then need a few sentences explaining how the 1D profiles can introduce artifacts.

Lines 286-289 are a little ambiguous. I'd clarify that the 1D profiles are what is plotted in Figure 3, and that the isosurface is used to show the shape of the Fe-rich liquids (but not to calculate a proxigram, which is what I thought at first).

Plagioclase – I would like to see more about the plagioclase. For example, line 122 suggests there is no Fe-enrichment in the plag. Which is a little surprising. But how do I see its absence in the figure 3? I don't think the profiles shown go into the plag.

At least six times in the paper (lines 70, 96, 109, 121, 137, 163) a trend is referred to, but instead of showing a figure, the reader is just referred to the online supplementary tables. I'd make plots of all of these and put them in the supplementary material.

A cartoon of how it is envisioned that the CBL grows during plagioclase growth, and how it interacts with the emulsion, would be helpful. Does it gobble up the Fe droplets, as some of the images suggest? If so, why do the droplets have a different composition? Could there be post-solidification diffusion of elements between the two glasses (re-equilibration), which would affect the small droplets more than the micron-scale CBL? Ideally would have a tip that cuts across the CBL into the emulsion.

How far does the nanoemulsion extend from the crystal face? The labeling in the images is vague. Maybe draw a boundary in the figures.

Line 161 – I thought the CBL was homogeneous. Do they have lateral compositional gradients?

High T solvus - Maybe the thermometer giving the high temperatures for immiscibility in the natural samples does not record the temperature at which the nanoemulsion formed. Since the thermometer is MgO based, it is controlled by growth of crystals. But maybe the emulsion formed after most crystallization occurred, at a lower T?

Comments on figures

Figure 1 – see above comments on labeling of various liquids

Figure 2 – A bit confusing how the locations of the different materials are indicated, with the 'Si-rich conjugate' on the bottom in a horizontal gray box, the 'Fe-rich boundary layer' on the side in a vertical box that also contains the numerical axis values for some of the oxides and 'PI' in a black box on the side that is dominantly labeled 'Oxide wt. % SiO₂, Al₂O₃'. Took me a little bit to figure

out what was going on. I'd suggest either removing all the boxes, and just have the x-axis labeled 'distance from Fe-rich boundary layer' or have a bar at the top with all three layers (as in figure 4), so the boxes don't have other text in them and are in the same orientation.

Also, why not show the composition of the Fe-rich boundary layer and PI on the same plot? That would make the plots look more like the inset cartoon. Took a little while to figure out how the inset related to the figure. Really, it is not exactly the same case, as there is no depleted melt layer on the right, just plag. Could leave the inset out, or make one that is closer to the plag-CBL-interstitial liquid that is being shown.

Figure 3 – Could use smaller steps in the 1D profile to smooth out the noise in the lines. Maybe 2-3 nm. Looks like 1nm or less was used. Left image, would like to see a profile across the plag-CBL boundary. Middle image – why not have the profile extend to the large Fe-rich conjugate at the tip of the needle? This would allow a better assessment of the compositional gradient in the Fe droplets. As it is, just includes a very small droplet. Right image – As stated above, I think you can see the effect of not using a proxigram here. The boundary between the Fe and Si-rich conjugates is thinner on the left than on the right. I think this is an artifact of the left boundary being closer to perpendicular with the 1D profile than the right boundary. I'm guessing this also affects the apparent thickness of the boundaries in the left and middle panels, as those boundaries are also curved.

Figure 4 – The text states that the CBLs were too thin (<5 microns) to get pure analyses of with the microprobe. So the CBL compositions are likely all have some amount of interstitial liquid and/or plag and/or fluorescence (Ca?) in them. So then not sure how much to trust the NBO/T calculation and whether the two liquids would mix linearly in NBO/T space since a number of elements go into that calculation. If the figure looks the same, I might just use Fe-content on the x-axis. Also, might at least comment on how the natural samples at 1020C match the experimental data quite well. Fortuitous?

Steve Parman

Reviewer #2 (Remarks to the Author):

The manuscript presents geochemical and textural data from a set of drillcore samples from the Kilauea Iki lava lake. The authors conclude from their electron probe data that small compositionally distinct rims of quenched liquid (CBLs) surrounding plagioclase represent the initial formation of Fe-rich liquid due to silicate liquid immiscibility. The manuscript also presents some of the first images of nano-scale heterogeneities in quenched basaltic glasses from Atom probe tomography, which they suggest shows early unmixing between Fe-rich and Si-rich liquids at the nm-scale.

In general, I do not think that the authors have made a clear case for their conclusions. In many places within the text, they treat their conclusions as a basis for assumptions (which is, at best, circular). Compositional boundary layers (CBLs) have been identified before surrounding phenocrysts in basaltic glass, but this is to my knowledge the first time that they have been attributed to small-scale features of liquid immiscibility. In most cases, previous work has suggested that they are very small disequilibrium features due to crystallization of the phenocryst.

Strangely, this manuscript suggests the CBL is disequilibrium as well but then attributes these minor features to larger-scale unmixing of the magma. I just don't see that they have provided the necessary support for this addition. In particular, I am not convinced that all of the geochemical signatures that they are seeing in the CBL are not just excluded elements from plagioclase crystallization instead of true small-scale silicate liquid immiscibility.

What is the primary evidence that this is definitively an immiscibility signature and not something

else? For example, the elements that commonly partition into the Si-rich conjugate of immiscible pair are the ones that are compatible in plagioclase and k-feldspar (e.g. Al, Na, K, Ba, Rb, etc.). The opposite is true for the Fe-rich conjugate. Plagioclase crystallization will exclude Fe, Ti, and HREE, and thus the melt immediately surrounding a growing plagioclase will be enriched in those elements.

My biggest concern with this manuscript is that the EPMA data collected are not reliable at the scale they need. In particular, the authors need to demonstrate that compositions are distinct on the < 1 micron scale (see their Fig.2), but they have used a 10 micron defocused beam for their measurements (see Methods). Even with a 0 micron point-source beam, at standard operating conditions of 10-15nA beam current, there is a ~5-10 micron secondary excitation volume whereby electrons from an adjacent mineral or glass will be activated and produce a "false" signal blurring the true contact. Figure 2 does not plot actual data points measured (only a smoothed line) so I am not sure where their measurements are actually from.

I think the authors have produced some interesting APT data, but I am not sure that they can make the conclusions presented here with the data as measured. I list other issues that should be addressed in revisions below:

L23 : this is stretching the impact of the contribution. I dont doubt that this is true, for other reasons, but I dont think that the two speculative sentences at the end of the paper warrant a mention in the abstract

L24: should state here explicitly what the Daly Gap is, including what compositions it ranges from.
L32-34: These implications are a bit of a stretch...

L57-58: please list in the main text what the range of Mg# and An contents of the phases are

L61-62: really? can you show this better with a reference to the supplementary material or a figure or table? As far as I can tell the method section basically says the opposite - that the CBL composition is not following the binodal because it is hard to analyze such small areas on the electron probe. So what are you basing this assertion on?

L65: perhaps list what the sub-solidus temperature is for this melt. Make sure to include uncertainties on all temperature estimates.

L73: this is a big leap all of the sudden. I dont' understand why some excluded elements around a crystal rim is now automatically an immiscible liquid. This is not justified so far.

L119: APT not defined yet

L142: Again, I believe that the CBLs are disequilibrium features, but I am just not sure that they are really Si-liquid immiscibility.

L148: I am not familiar with this technique - can you explain it in more detail or at least provide a reference for this calculation. What are the uncertainties on the calculation?

L156-157: so are you suggesting that you amazingly captured the crystal in the act of growing an albite rim from an otherwise homogenous core? otherwise, I would imagine that the formation of a CBL is a nearly constant process that then gets erased as the system progresses to and past the solidus.

L162-166: these are all really long complicated sentences. they should really be broken up.

L171: how is that possible? Samples that are only 3m apart are over 100°C different from one another, but now we are supposed to believe that this T gradient persisted for several years?

L230-232: this seems like a lot of speculation that isn't founded by the data or results presented. if anything, what they are claiming is directly contradicted by the sentence immediately prior which actually discusses the data. If the T is high, then the immiscibility is near the top of the binodal, and the compositional difference (and hence density difference) will be small.

L259-261: so then are the data in fig 2 averages of these profiles???? especially with a defocused 10 um beam, you can't possibly have correct spatial data plotted in fig 2

Figures: Figure 2 is not well constructed. How are the two Y-axes labeled? Is it oxide wt% of SiO₂ and Al₂O₃ in plagioclase on the right side? what does that mean? label the inset better. what does the up and down arrows mean in the inset? is that the same for all elements? what are the actual data points? there is no way that they have an analysis every 1 um for example. maybe every 5 microns? for EPMA analysis you have a secondary excitation volume, which means that usually for 10nA beam current, your secondary excitation volume is about ±5 microns. 5 microns is exactly what they are seeing in terms of an increase in Al₂O₃ and a decrease in FeO - which could just be from getting the beam close to the plagioclase grain. I am not sure that I buy that these data are correctly recording the glass composition and not any other analytical artifacts in this figure. At the very least they should put data points on the figure and not just a smooth line.

Reviewer #3 (Remarks to the Author):

I think the data and discussions in this paper make an interesting and important addition to previous work on silicate liquid immiscibility in basaltic magmas. However, I think it is a stretch (unjustified) to say this extension would "conceivably facilitate the downward movement of this dense inviscid liquid " in a natural tholeiitic system. I will admit that it could if the same CLB development described here could be shown to occur in a much slower cooling environment where crystal nucleation and growth rates and diffusion might yield a different crystal-melt product. It might be helpful to see an estimate of what the total crystallinity of the Hawaii samples would be for each of the samples studied in the 1112 to 1017 C temperature range assuming no dissolved water effect. I am assuming, using experimental data (e.g., Dixon and Rutherford, 1979) that the original basalt is 65-80 percent crystallized, but you could make a better determination. The problem that you, and all of us have, is convincing the rest of the science community that separation of the Fe-rich melt (or crystalline equivalent) from the Si-rich melt is possible at these or even higher magma crystallinities. The fact that the build-up of water with crystallization in such a basalt cooling at depth or ascending to lower depths causes earlier crystallization of Fe-oxides (Spulber and Rutherford, 1983, J Petrol) doesn't help make the case for SLI as an important process at depth.

In conclusion, the paper can be a valuable addition to the work that has been done on silicate liquid immiscibility based on the new data presented which shows an interesting connection between melt boundary layers on different phases and a developing immiscibility in the residual melt. The discussion and explanation of these very fine scale features is a good compliment to the analyses. However, the extrapolation of the significance of this data to things such as the Daley Gap is not justified, and thus the significance of the paper is not as large as would appear from statements in the introduction and conclusion sections.

Malcolm J Rutherford

REVIEWERS' COMMENTS

REVIEWER #1 (REMARKS TO THE AUTHOR):

General comments – This paper presents new micro and nano analyses relating to liquid immiscibility in crystal-rich basaltic rocks from three localities: Hawaii, Iceland and the Snake River Plain. The authors find that plagioclase crystals in some samples are surrounded by an Fe-rich compositional boundary layer (CBL) up to a few microns thick. SEM and atom probe imaging of the glass outside of these CBLs is a nanoemulsion of Fe-rich droplets in a Si-rich melt. The conclusion is that the CBL formed due to fast crystallization of the plagioclase and that the ensuing Fe-enrichment produced liquid immiscibility that formed the nanoemulsion. The analyses include the first application of atom probe tomography (a relatively new analytical method) to immiscibility in basalts. The paper is convincing that liquid immiscibility occurred in these samples by the mechanism proposed (crystal growth and spinodal decomposition).

We are happy to learn that the reviewer agrees with us that the subject of our study is important and our paper is convincing.

1. How widespread this process is, and how it relates to fractionation of basalts and the 'Daly gap' is another question. Such extreme CBLs appear to be pretty rare. The CBLs in the samples are up to a few microns width and so would be observable with a standard microprobe (Figure 2a) if they were present in a sample. I think people would have seen them in more localities if they were common. I've not seen them. So not sure that the presence of CBLs in these three samples convinces me this is a widespread process.

Regarding the question of how widespread the CBLs are, in this contribution we show CBLs from three localities from differing tholeiitic provinces: Hawaii, Iceland and Snake River Plain. Additionally we collected further data on the Fe-rich CBLs from another drill core from the Kilauea Iki lava lake, showing the phenomenon. Similar Fe-rich CBL rims around plagioclase have also been observed in samples from the Galapagos spreading ridge (pers. comm. Matthew Gleeson). Furthermore, based on the fact that the samples from the studied localities have all been previously characterised (Helz 1980; Helz 1987; Helz et al. 2014; Jean et al. 2018; Neave et al. 2017; Neave et al. 2013) and no CBLs have been reported, leads us to suggest that these CBLs can easily be overlooked unless the scientist is specifically looking for such features. It is clear that depending on the contrast-brightness optimisation for BSE imaging, and the magnification used to examine the samples, such features can be easily overlooked. We now add a sentence to highlight this potential to overlook CBLs in the discussion (line 61 and 240).

2. Also, the CBLs are very thin, don't show a lot of evidence for migrating and appear to only form when the samples are highly crystalline. So how they would ever migrate to form large enough magma bodies to explain the Daly gap is also not clear.

It should be noted that this manuscript focuses on the mechanism of immiscible liquids forming CBLs around growing crystals. We state this in the introduction (line 27) and in the discussion (line 270 and 283). To further clarify the focus of the manuscript, we have added a new figure (Fig. 4), to schematically illustrate the emulsion evolution. We retain a tentative statement in the discussion, eluding to the fact that our findings could help us understand the Daly gap (line 292); however, we remove reference to the Daly Gap from the abstract.

3. So, I think the study shows an example of interesting thermodynamic (spinodal decomposition) and kinetic (CBL) effects, but I am unconvinced of their connection to larger magmatic processes. Maybe the CBLs or emulsions can be used to extract cooling rates in magmas where they are found?

We maintain that the production of the CBLs may be significant in our search for the formation of Fe-rich and Fe-poor silicate melts. However, we have toned-down this connection (see comment 2) as we cannot prove with our sample set (that only show the very start of the process) that the melts migrate.

Extraction of cooling rates: this is an interesting question and would be an intriguing tool to the petrologist. Unfortunately, we do not believe extracting an absolute cooling rate from the thickness of the CBL rims is possible from our data. Our findings show that crystal growth rate, the crystal face in question, crystal morphology, time, and temperature all play an important role in the development of the CBL. Currently, we cannot look at these factors independently. In order to do so would require an in-depth numerical approach that focuses on establishing a robust link between the CBLs and cooling rate. This is a whole new project and cannot be part of this work. However, without doubt, establishing this link would be a worthwhile pathway for the future. Thank you for the suggestion.

Specific comments -

4. Terminology - The terminology should be simplified and used more consistently in the text and figures. For example, in the text, the liquid layer around the crystals is called a CBL (compositional boundary layer), but in Figure 1, it is labeled 'Fe-rich boundary layer', which is not the same as the 'Fe-rich conjugate' found in the emulsion. Similarly, the glass around the crystals (and CBL) is referred to as 'interstitial liquid' in some places in the text (line 124) but labeled as 'Si-rich conjugate' in some panels, and 'nanoemulsion' in other panels. And that is not to be confused with the 'Si-rich boundary layer', which only occurs around the pyroxenes. I suggest trying to simplify the terminology. For the compositional layers around the crystals, maybe try to consistently use the abbreviation CBL, as in text. Could use 'Fe-rich CBL' for the plagioclases and 'Si-rich CBL' for the pyroxenes in the figures. I would drop the use of 'Si-rich conjugate' as it suggests it is different than the interstitial liquid. I suggest using 'interstitial liquid' to refer to the glass outside the CBL, including the labeling in the images. For example, in Figure 2, 'Si conjugate' would be replaced by 'interstitial liquid'. And you can just refer to the compositional gradients in the interstitial liquid that were produced by the formation of the CBL. It is not clear that the formation of the CBL or the nanoemulsion significantly increased the Si in the interstitial liquid. I don't see it in figures 2 or 3. For the Fe-rich immiscible liquid droplets, could use Fe-rich conjugate. But might be clearer to a wider audience if you call them immiscible liquids, which is what most people think of. So something like 'Fe-rich immiscible droplets'.

We have changed the manuscript, figures, and supplementary figure to simplify the terminology using the very useful suggestions above.

Fe-rich CBL = Fe-rich glass surrounding plagioclase.

Si-rich CBL = Si-rich glass surrounding pyroxene.

Interstitial glass = glass outside of the CBL. If a nanoemulsion is observable within the interstitial glass it is stated.

Fe-rich immiscible droplet = Fe-rich glass with a droplet morphology.

5. APT analysis – APT analysis is pretty new for geoscience. So exactly how to present the data, which included millions of data, is still being worked out. At some level, ideally would want the RHIT and range files, and the shank angle for the reconstruction. That's what you would need to reproduce, for example, Figure 3. Given that most people don't have IVAS, maybe that can be an 'upon request' deal.

We have added additional information to the methods section on APT reconstruction (line 349-365): we now provide information regarding the shank reconstruction and the shape of the needles (consistent taper angles, ~10 degrees). For the samples we used the SEM derived shape to calibrate the reconstruction, we used a 10 degree shank angle to provide an initial reconstruction. These reconstruction parameters were subsequently used for all reconstructions in this report.

We are happy to provide the raw data files 'upon request' and have said so in the supplementary figure 17.

6. For the interested researcher, I think giving the voltage curve for the analysis would be good. It tells you a lot about how well things went during evaporation.

We now include a voltage curve for sample 173770 at 14 μm from the Fe-rich CBL in the supplementary figure 16. The figure represents voltage evolution typical for the many analyses collected as part of this report. Approximately 20.4 million ions were collected before specimen failure (indicated by the sharp voltage drop at the end of the plot and denoted by the dashed blue line). Only ions acquired after sufficiently stable ion evaporation and before failure (~0.5-20.0 million ions) were used for reconstruction and subsequent chemical analysis. The voltage trends were generally smooth, indicating stable analysis. The general gentle increase in voltage is related to the increase in the cross-section of the atom probe tip (as shown by the figure now included in the supplementary figure 16). However, there were occasional voltage drops during analysis, e.g. at 9.3, 13.8, and 14.9 million ions (red dashed lines). These voltage drops are the instrument's response to increases in detected ion rates. For example, the evaporation at 14.9 million ions briefly went from 1.0 +/- 0.3% to ~7% causing the instrument to respond. The control algorithm adjusts voltage to maintain a target, and achieve a constant detected ion rate. The fact that the voltage spikes are not related to the compositional changes (supplementary figure 16) makes us confident that our analysis is robust.

In general, significant ionization spikes may indicate uncontrolled removal of specimen material suggesting that either some small depth of material has been removed but not fully detected (resulting in a discontinuity in the analysed structure), or just a normal response due to normal evaporation variations. Cases where significant losses of collected material have occurred are indicated when the previous, longer-scale voltage trends are not resumed (a clear voltage discontinuity) and these regions were filtered out and not used as continuous volumes as part of this study. For cases like those shown in supplementary figure 16, where the voltage trends were resumed, the reconstructed volumes are deemed robust. Only analysis identified as robust are used in this study.

7. And the mass/charge spectra. For example, the interstitial glass seems to have about 12% FeO (Figure 2), but the Fe concentration in the left panel of Figure 3 drops to almost zero in the interstitial glass. What mass/charge range was used for this profile? Fe+1? Fe+2? FeO+1? Fe2O3? Would be useful to know. So I'd suggest including those, as well as the length of the needles, the tip

radius and shank angle. This will help future researchers wanting to do similar analyses decide how to make their tips.

We have included a figure of the mass/charge spectra in the supplementary figure 17, with additional information. We used Fe+1 and Fe+2 (see supplementary figure 17).

We have added additional information to the methods section on APT reconstruction (line 358).

8. Ideally, would use proxigrams to get the compositional profiles across the boundaries, rather than 1D profiles. The difference is that a proxigram can be used to calculate the compositional gradient perpendicular to the surface at all points. This should give the most accurate profile. The 1D profile does not do this, and just averages flat slices. If the boundary is curved, or not orthogonal to the direction of the 1D profile, this will artificially smear out the boundary. I think you can see this effect in the right panel of figure 3. The boundary between the Fe and Si-rich conjugates is thinner on the left than on the right, because the right boundary is more tilted with respect to the 1D profile direction. If proxigrams aren't used, then need a few sentences explaining how the 1D profiles can introduce artefacts.

We understand the concern of the reviewer, in fact, to begin with we used proxigrams to get the compositional profiles across the boundaries – based on the same line of thinking as presented by the reviewer. However, we realised that because of the complicated 3D interface, the proxigram data was significantly influenced by artefacts from two or three adjacent Fe-rich immiscible phases – i.e. there was a “smearing effect”. We therefore decided to look at quasi 1D profiles, so that we could isolate areas of the emulsion interface that were perpendicular to the 1D profile. The 1D profile is not truly 1D: it is an elongate bar with a square profile of 10 nm by 10 nm and the length as shown on the profiles in Fig. 3. This way we could chose a 1D profile that is perpendicular to the interface of the Fe-rich and Si-rich nanoemulsion. We now clarify the specifics of the 1D profile and the reason behind using it in the methods section (line 366-375).

9. Lines 286-289 are a little ambiguous. I'd clarify that the 1D profiles are what is plotted in Figure 3, and that the isosurface is used to show the shape of the Fe-rich liquids (but not to calculate a proxigram, which is what I thought at first).

We have now added this clarification to the manuscript, under the section headed: “APT measurements and analysis” (line 366-375) – see comment 8.

10. Plagioclase – I would like to see more about the plagioclase. For example, line 122 suggests there is no Fe-enrichment in the plag. Which is a little surprising. But how do I see its absence in the figure 3? I don't think the profiles shown go into the plag.

From the APT data, 1D profiles cross-cutting the plagioclase-CBL interface show that there is negligible Fe in the plagioclase (Fig. 3). Furthermore, there is no variation of the Fe across the plagioclase (on the scale analysed – 250 nm). We have added figures to the supplementary material showing the results of the APT across a plagioclase for an Fe, Al, and (Ca/Ca+Na+K)-transect (Supplementary figure 11-12). We also add in major element EPMA data to table 1 for plagioclase rims and refer to this additional data in the text (lines 57).

11. At least six times in the paper (lines 70, 96, 109, 121, 137, 163) a trend is referred to, but instead of showing a figure, the reader is just referred to the online supplementary tables. I'd make plots of all of these and put them in the supplementary material.

Plots of the trends referred to in the text have now been added as supplementary figure.

CBL thickness vs rim AR (supplementary figure 6)

CBL thickness vs depth (supplementary figure 8)

FeO wt. % of the interstitial liquid vs depth in the drill core (supplementary figure 9)

Al and Na concentration of the Fe-rich CBL, plagioclase, and Fe-rich immiscible liquid of the nanoemulsion (supplementary figure 13)

Homogeneity of the Fe-rich CBL (supplementary figure 11)

CBL contact with plagioclase (supplementary figure 12)

CBL contact with interstitial liquid (shown in Fig. 3 of the manuscript – due to the difference in composition, the glass tips regularly broke across the boundary between the Fe-rich CBL and interstitial glass)

12. A cartoon of how it is envisioned that the CBL grows during plagioclase growth, and how it interacts with the emulsion, would be helpful. Does it gobble up the Fe droplets, as some of the images suggest? If so, why do the droplets have a different composition? Could there be post-solidification diffusion of elements between the two glasses (re-equilibration), which would affect the small droplets more than the micron-scale CBL?

We agree that a cartoon would be very useful and have now constructed one (Fig. 4). We now changed the text to refer to the different parts of the figure explicitly.

Different composition of droplets: The Fe-rich droplets have higher Fe concentrations at shallower levels in the Kilauea (see table 1). Within a single sample the Fe-rich droplets typically have variations in concentrations below 1 standard deviation (see table 1). The size of the probe beam interaction volume and the effect on the Fe-rich droplet analyses is discussed in the Methods section (line 307).

Post-solidification diffusion of elements? We think that there is very little post-solidification of elements as we do not see typical diffusion related elemental profiles in our 1D profiles (figure 3). This is now explicitly mentioned in the text (line 152).

13. Ideally would have a tip that cuts across the CBL into the emulsion.

In fact, there is one glass APT tip (sample: 179377) that crosses the CBL-interstitial liquid interface (Figure 3). The interstitial liquid is comprised of a nanoemulsion. We now explicitly refer to this tip in the text (line 160). It is not possible to cut a glass tip long enough to intersect all the CBL and a significant amount of interstitial liquid.

14. How far does the nanoemulsion extend from the crystal face? The labeling in the images is vague. Maybe draw a boundary in the figures.

At the scale of the BSE imaging, the nanoemulsion is observed to fine away from the crystal face, as mentioned in the manuscript (line 96). As the nanoemulsion fines, it becomes too small to resolve by SEM (e.g. Fig. 1b) at magnification greater than a half field width of 50 μm . The APT measurements extend 14 μm away from the crystal edge and the nanoemulsion remains present at that scale. As such, without further APT measurements, a definite boundary for the nanoemulsion cannot be

determined. Judging from BSE imaging, it is predicted that it may vary between locality samples. We now state in the methods that BSE imaging is at its limits (line 300) and APT analysis showed that the interstitial liquid comprises a nanoemulsion up to 14 μm from the crystal contact (line 157 and 371).

Note: The process of nanoemulsion formation is discussed (lines 226-230), proposing different hypotheses as to how it could form.

15. Line 161 – I thought the CBL was homogeneous. Do they have lateral compositional gradients?

From the APT data, 1D profiles across the CBL show that there are no compositional gradients (on the scale analysed – less than 200 nm). Supplementary figure 11 shows the homogeneity of the Fe-rich CBL; this supplements the profile shown in Fig. 3. We realize that these facts were not made explicitly clear. This is now corrected in the text (line 146)

16. High T solvus - Maybe the thermometer giving the high temperatures for immiscibility in the natural samples does not record the temperature at which the nanoemulsion formed. Since the thermometer is MgO based, it is controlled by growth of crystals. But maybe the emulsion formed after most crystallization occurred, at a lower T?

Lines 70-71 clarify the temporal nature of the cooling rate experienced by the core samples from the Kilauea Iki lava lake: there is a downward migration over time of the lava lake thermal maximum. It is also important to note that there is a distinction to be made between the quench (during drilling) and the accelerated cooling in samples closer to the (temporally expanding) lava lake crust. We have studied additional samples from drill core from 1979, 1981, and 1988. These all show Fe-rich CBLs around plagioclase of varying thickness. We have quantified observations of the Fe-rich CBL thickness from the 1981 drill core to supplement the existing data. This data shows the same trends as the 1976 drill core. This is now presented in table 2 and discussed in line 118-120.

Comments on figures

17. Figure 1 – see above comments on labelling of various liquids

We have changed the manuscript to simplify the terminology, using the very useful suggestions given above (see reply to comment number 4).

18. Figure 2 – A bit confusing how the locations of the different materials are indicated, with the ‘Si-rich conjugate’ on the bottom in a horizontal gray box, the ‘Fe-rich boundary layer’ on the side in a vertical box that also contains the numerical axis values for some of the oxides and ‘Pl’ in a black box on the side that is dominantly labeled ‘Oxide wt. % SiO₂, Al₂O₃’. Took me a little bit to figure out what was going on. I’d suggest either removing all the boxes, and just have the x-axis labeled ‘distance from Fe-rich boundary layer’ or have a bar at the top with all three layers (as in figure 4), so the boxes don’t have other text in them and are in the same orientation.

As suggested, we have removed the surrounding boxes and have relabelled the x-axis “Distance from Fe-rich CBL”.

19. Also, why not show the composition of the Fe-rich boundary layer and Pl on the same plot? That would make the plots look more like the inset cartoon. Took a little while to figure out how the inset related to the figure. Really, it is not exactly the same case, as there is no depleted melt layer on the

right, just plag. Could leave the inset out, or make one that is closer to the plag-CBL-interstitial liquid that is being shown.

As suggested, we have added a representative Fe-rich CBL and plagioclase rim analysis to the figure and removed the inset. Note that new mineral compositions have also been added to table 1.

20. Figure 3 – Could use smaller steps in the 1D profile to smooth out the noise in the lines. Maybe 2-3 nm. Looks like 1nm or less was used.

We are at the spatial resolution limit of the technique, we cannot go smaller. Since we do not want to artificially smooth the data, but rather show the real data, we have kept the figure as it was in this respect.

21. Left image, would like to see a profile across the plag-CBL boundary.

We have now added such a profile to Fig 3 and changed the text and figure caption accordingly.

22. Middle image – why not have the profile extend to the large Fe-rich conjugate at the tip of the needle? This would allow a better assessment of the compositional gradient in the Fe droplets. As it is, just includes a very small droplet.

The large Fe-rich conjugate does not have a perpendicular intersection with a 1D compositional profile, therefore there is a smearing effect and the gradient recorded is an artefact. To help the reader, we now discuss this geometric effect on the apparent slope of the chemical profile in the methods section (lines 372-375 and comment 8).

23. Right image – As stated above, I think you can see the effect of not using a proxigram here. The boundary between the Fe and Si-rich conjugates is thinner on the left than on the right. I think this is an artefact of the left boundary being closer to perpendicular with the 1D profile than the right boundary. I'm guessing this also affects the apparent thickness of the boundaries in the left and middle panels, as those boundaries are also curved.

See comment 22.

24. Figure 4 – The text states that the CBLs were too thin (<5 microns) to get pure analyses of with the microprobe. So the CBL compositions are likely all have some amount of interstitial liquid and/or plag and/or fluorescence (Ca?) in them. So then not sure how much to trust the NBO/T calculation and whether the two liquids would mix linearly in NBO/T space since a number of elements go into that calculation. If the figure looks the same, I might just use Fe-content on the x-axis.

The figure does look the same for Fe-content on the x-axis, this has been added as supplementary figure 14. We agree that the CBL compositions inevitably have some degree of mixing, which is why we take the most extreme (i.e. the end member) numbers to define the binodal. We discuss the consideration taken to address this analytical resolution issue in the Methods section (line 314). Silicate liquid immiscibility is best thought of using a combination of the elements which preferentially partition into each conjugate, therefore we have kept Fig. 5 as NBO/T: it better reflects all the elements involved in the silicate liquid immiscibility, rather than concentrating on just Fe.

25. Also, might at least comment on how the natural samples at 1020C match the experimental data quite well. Fortuitous?

This was mentioned, but we have clarified the manuscript (line 224, 254) to include the temperature of interest (1020°C) and added in a reference to the figure so readers can observe the match.

Steve Parman

REVIEWER #2 (REMARKS TO THE AUTHOR):

The manuscript presents geochemical and textural data from a set of drillcore samples from the Kilauea Iki lava lake. The authors conclude from their electron probe data that small compositionally distinct rims of quenched liquid (CBLs) surrounding plagioclase represent the initial formation of Fe-rich liquid due to silicate liquid immiscibility. The manuscript also presents some of the first images of nano-scale heterogeneities in quenched basaltic glasses from Atom probe tomography, which they suggest shows early unmixing between Fe-rich and Si-rich liquids at the nm-scale. In general, I do not think that the authors have made a clear case for their conclusions. In many places within the text, they treat their conclusions as a basis for assumptions (which is, at best, circular).

Compositional boundary layers (CBLs) have been identified before surrounding phenocrysts in basaltic glass, but this is to my knowledge the first time that they have been attributed to small-scale features of liquid immiscibility. In most cases, previous work has suggested that they are very small disequilibrium features due to crystallization of the phenocryst. Strangely, this manuscript suggests the CBL is disequilibrium as well but then attributes these minor features to larger-scale unmixing of the magma.

26. I just don't see that they have provided the necessary support for this addition. In particular, I am not convinced that all of the geochemical signatures that they are seeing in the CBL are not just excluded elements from plagioclase crystallization instead of true small-scale silicate liquid immiscibility. What is the primary evidence that this is definitively an immiscibility signature and not something else? For example, the elements that commonly partition into the Si-rich conjugate of immiscible pair are the ones that are compatible in plagioclase and k-feldspar (e.g. Al, Na, K, Ba, Rb, etc.). The opposite is true for the Fe-rich conjugate. Plagioclase crystallization will exclude Fe, Ti, and HREE, and thus the melt immediately surrounding a growing plagioclase will be enriched in those elements.

Liquid immiscibility is defined as “the unmixing of magmas into liquids of contrasting composition” (Philpotts and Ague 2009). An Fe-rich immiscible liquid has a composition of approximately pyroxene composition and closely matches the composition of the Fe-rich conjugate in the system fayalite-leucite-silica. This has been shown by a plot of the composition of immiscible liquids preserved as glassy droplets in a range of tholeiitic volcanic rocks, experiments and melt inclusions, with our new data added on top – this figure is added as supplementary figure 5. This manuscript investigates the process by which unmixing can happen. Excluded elements from the plagioclase crystallisation have formed an unmixed liquid – as very clearly seen in the BSE SEM images. All this is now shown clearly in the new conceptual summary figure (Fig. 4).

27. My biggest concern with this manuscript is that the EPMA data collected are not reliable at the scale they need. In particular, the authors need to demonstrate that compositions are distinct on the < 1 micron scale (see their Fig.2), but they have used a 10 micron defocused beam for their measurements (see Methods). Even with a 0 micron point-source beam, at standard operating

conditions of 10-15nA beam current, there is a ~5-10 micron secondary excitation volume whereby electrons from an adjacent mineral or glass will be activated and produce a "false" signal blurring the true contact. Figure 2 does not plot actual data points measured (only a smoothed line) so I am not sure where their measurements are actually from.

The effect of secondary excitation was considered and modelled. Using an end-member Fe-rich glass composition, a 2 μm beam diameter, and 15 kV, we simulated the interaction volume of the beam using Casino v2.48. Our modelling shows that less than 25% of the energy of the beam extends beyond 1.4 μm from the central point of a 2 μm diameter beam (supplementary figure 15). While activation of electrons from an adjacent mineral or glass is a concern we share with the reviewer, the effect observed in the data is the reverse of what would be expected were we getting a "false" signal. From the SEM BSE signal, we can confidently say that the brighter areas correspond to regions of the sample where the average atomic number is higher. SEM EDS and APT analyses show that the Fe-rich CBL is enriched in Fe compared to the plagioclase and surrounding interstitial glass. Therefore if we were mixing the interstitial glass with an adjacent phase, we would expect to see a smearing effect that enriches our analyses in elements with a high atomic number e.g. Fe, Mn. We see the opposite, FeO decreases towards the Fe-rich CBL. Such a concentration profile is well documented in the nucleation and growth of a two-phase mixture in the alloy material sciences literature, see Fig 2 of Findik (2012) (see image below).

The reviewer rightly pointed out a 10 μm diffuse beam was stated in the methods, which clearly is greater than the spacing of the samples. On re-checking the beam parameters used, the 10 μm defocused beam was not used for the glasses. It was a 2 μm beam and this mistake has been corrected in the manuscript (line 307). Please accept our apologies for this mistake in the original submission. To check the veracity of using a more focused beam on glass, 40 analyses of well-characterised natural and synthetic glasses (Basalt glass Makaopuhi Lava Lake HI NMNH 113498-1 A-99; and Corning Glass Reference 'D' NMNH 117218-3) were made using a 2 μm and 10 μm beam diameter (20 analyses with a 2 μm beam and 20 with a 10 μm beam). The total dataset of 40 analyses (regardless of beam size) had a standard deviation of 0.5 wt.% or less for all elements (Honour et al. in review).

The data points have been added to figure 2.

28. I think the authors have produced some interesting APT data, but I am not sure that they can

make the conclusions presented here with the data as measured. I list other issues that should be addressed in revisions below:

This study tackles a previously overlooked phenomenon of rims surrounding plagioclase (as seen in BSE images) and shows that magma unmixing can be promoted by growing crystals. This study draws on a wide range of disciplines from Earth Sciences and Material Sciences. We combine a range of data sets, including a significant number of atom probe tomography samples, electron microprobe work, and image analysis from four different sample suites. Consequently, we strongly disagree that we have insufficient data measured.

29. L23 : this is stretching the impact of the contribution. I don't doubt that this is true, for other reasons, but I don't think that the two speculative sentences at the end of the paper warrant a mention in the abstract

We have now removed the potential larger impact from the abstract.

30. L24: should state here explicitly what the Daly Gap is, including what compositions it ranges from.

The manuscript focuses on the mechanism of immiscible liquids forming CBLs around crystals. See comment 29. We retain a tentative statement in the discussion, alluding to the fact that our findings could help us understand the Daly gap (line 292); however, more work at different sample and resolution scales would be needed to confirm this link.

31. L32-34: These implications are a bit of a stretch...

We have toned down the larger impact and reworded this line to say "we may be able to better understand the origin".

32. L57-58: please list in the main text what the range of Mg# and An contents of the phases are

The compositional evolution of the glass is expressed by Fig. 5 using the measure of NBO/T rather than Mg# calculated from specific mineral chemistry. The mineral chemistry for the rims of olivine, pyroxene and plagioclase have been measured by EPMA and we have now added this data to table 1 and referred to it in line 57, also comment 10.

33. L61-62: really? can you show this better with a reference to the supplementary material or a figure or table? As far as I can tell the method section basically says the opposite - that the CBL composition is not following the binodal because it is hard to analyze such small areas on the electron probe. So what are you basing this assertion on?

See reply to comment 26. The newly drawn binodal shown in Fig. 5 is defined by the end-member CBL compositions, as stated in lines 246-248 and 312-315.

34. L65: perhaps list what the sub-solidus temperature is for this melt. Make sure to include uncertainties on all temperature estimates.

It is only possible to determine the solidus temperature. The solidus in Kilauea Iki occurs at roughly 980 °C, but that is a little tricky, as the end-stage rhyolitic glass persists metastably for a bit before finally disappearing.

As for uncertainties in the temperature measurements, Helz & Thornber report $\pm 8^\circ$ absolute for that calibration. The reproducibility (aka internal precision) of the MgO and CaO determinations is more like ± 3 degrees (Helz & Hearn, 1998). References to uncertainties have been added to the manuscript (line 68-69).

35. L73: this is a big leap all of the sudden. I don't understand why some excluded elements around a crystal rim is now automatically an immiscible liquid. This is not justified so far.

See reply to comment 26, and new supplementary figure 5 comparing the composition of the Fe-rich glasses to known immiscible liquids from volcanic rocks, melt inclusions, and experiments.

36. L119: APT not defined yet

Defined on line 34.

37. L142: Again, I believe that the CBLs are disequilibrium features, but I am just not sure that they are really Si-liquid immiscibility.

See reply to comment 26, 35, and see the added supplementary figure 5.

38. L148: I am not familiar with this technique - can you explain it in more detail or at least provide a reference for this calculation. What are the uncertainties on the calculation?

The method for calculating plagioclase growth rates has been added to Methods section, with a discussion of potential uncertainties on the calculation (line 327-340).

39. L156-157: so are you suggesting that you amazingly captured the crystal in the act of growing an albite rim from an otherwise homogenous core? otherwise, I would imagine that the formation of a CBL is a nearly constant process that then gets erased as the system progresses to and past the solidus.

The process of CBL formation is not necessarily a constant process; it would require the necessary composition of the interstitial liquid and a constant plagioclase growth rate that exceeds the diffusion rate of elements in the interstitial liquid. Given the varying cooling rate of the Kilauea lava lake (as a result of the thermal regime), it is possible that the plagioclase growth rate is not constant and therefore CBL formation is not a constant process. However, we do agree with the reviewer that once past the solidus, there is no record of the CBL. We state this at the beginning of the manuscript, as a reason why the samples we use are suitable for this study (as they provide such a fantastic example of this phenomenon, quenched in-situ – line 37).

40. L162-166: these are all really long complicated sentences. they should really be broken up.

We have broken these up and rephrased to aid clarity.

41. L171: how is that possible? Samples that are only 3m apart are over 100°C different from one another, but now we are supposed to believe that this T gradient persisted for several years?

This particular gradient can be found in drill core recovered over more than a decade, but the thermal gradient moves. If the same location had been drilled 6 months earlier, the isotherms for

the samples would have been about 1 metre shallower than they are. If 6 months later, they would have been about a metre deeper. We have addressed the idea of temporal variation in the manuscript to clarify this (lines 70-72). A more in-depth discussion on this can be found in Helz et al (2014).

42. L230-232: this seems like a lot of speculation that isn't founded by the data or results presented. if anything, what they are claiming is directly contradicted by the sentence immediately prior which actually discusses the data. If the T is high, then the immiscibility is near the top of the binodal, and the compositional difference (and hence density difference) will be small.

We agree that the density differences calculated from the glass compositions at high temperatures make the gravitationally-driven separation of two immiscible liquids unlikely. We have clarified our discussion on line 281-282.

43. L259-261: so then are the data in fig 2 averages of these profiles???? especially with a defocused 10 um beam, you can't possibly have correct spatial data plotted in fig 2

See comment 27. It is important to note that figure 2 plots a transect from the interstitial liquid towards an Fe-rich CBL surrounding a plagioclase. The lines highlighted by the reviewer are discussing measurements of the interstitial glass and the Fe-rich immiscible droplets.

44. Figures: Figure 2 is not well constructed. How are the two Y-axes labeled? Is it oxide wt% of SiO₂ and Al₂O₃ in plagioclase on the right side? what does that mean?

Figure 2 has been relabelled to ensure the x-axis and y-axis are very clear and easy to read.

45. label the inset better. what does the up and down arrows mean in the inset? is that the same for all elements?

We have removed the inset – we realise it was adding confusion to figure 2 and was not adding to the argument. We do, however, use the same concept that was illustrated in the inset, in our reply to comment 27.

46. what are the actual data points? there is no way that they have an analysis every 1 um for example. maybe every 5 microns? for EPMA analysis you have a secondary excitation volume, which means that usually for 10nA beam current, your secondary excitation volume is about ±5 microns. 5 microns is exactly what they are seeing in terms of an increase in Al₂O₃ and a decrease in FeO - which could just be from getting the beam close to the plagioclase grain.

See comment 27.

47. I am not sure that I buy that these data are correctly recording the glass composition and not any other analytical artifacts in this figure. At the very least they should put data points on the figure and not just a smooth line.

See comment 27. Data points have been added to the figure.

REVIEWER #3 (REMARKS TO THE AUTHOR):

Particular editorial/grammatical and minor comments/suggestions annotated on the manuscript were very helpful and have been added to the revised manuscript.

I think the data and discussions in this paper make an interesting and important addition to previous work on silicate liquid immiscibility in basaltic magmas.

48. However, I think it is a stretch (unjustified) to say this extension would "conceivably facilitate the downward movement of this dense inviscid liquid " in a natural tholeiitic system. I will admit that it could if the same CLB development described here could be shown to occur in a much slower cooling environment where crystal nucleation and growth rates and diffusion might yield a different crystal-melt product.

We agree that migration of any Fe-rich liquid is a trade-off between coarsening of the Fe-rich liquid and a narrowing of pore-throats as crystallinity increases. The accumulated Fe-rich liquid surrounding the plagioclase grains is coarser than an emulsion of suspended Fe-rich droplets and so has a greater propensity to migrate. However, we do not observe evidence of such migration. In our samples, given the density differences, this is not observed. So the phrase is qualified by a preceding comment regarding density evolution (line 281-282).

49. It might be helpful to see an estimate of what the total crystallinity of the Hawaii samples would be for each of the samples studied in the 1112 to 1017 C temperature range assuming no dissolved water effect. I am assuming, using experimental data (e.g., Dixon and Rutherford, 1979) that the original basalt is 65-80 percent crystallized, but you could make a better determination.

Melt percentages for different samples from the 1976 drill core have been added to the manuscript, following Barth et al. (1994) (line 57).

50. The problem that you, and all of us have, is convincing the rest of the science community that separation of the Fe-rich melt (or crystalline equivalent) from the Si-rich melt is possible at these or even higher magma crystallinities.

See reply to comment 48.

51. The fact that the build-up of water with crystallization in such a basalt cooling at depth or ascending to lower depths causes earlier crystallization of Fe-oxides (Spulber and Rutherford, 1983, J Petrol) doesn't help make the case for SLI as an important process at depth.

The Kilauea Iki lava lake is an open system with respect to H₂O, as evidenced by the presence in the partially molten mush of a sparse population of (very low pressure) steam bubbles. All glassy samples have some bubbles in the glass, but the bubbles are just passing through, and the confining pressure is mostly < 6 bars. We see no reason to expect a significant effect on the CBLs. This is explained in the manuscript (line 133-135).

Our observations in Hawaii, Snake River Plain and Laki are for low pressure systems. In crustal magma chambers deeper than our system, previous work has suggested that H₂O enrichment during differentiation promotes magnetite crystallisation and therefore Fe depletion; however, recent work by Hou et al, 2018, suggests that hydrous systems may also reach silicate liquid immiscibility.

In conclusion, the paper can be a valuable addition to the work that has been done on silicate liquid immiscibility based on the new data presented which shows an interesting connection between melt boundary layers on different phases and a developing immiscibility in the residual melt. The

discussion and explanation of these very fine scale features is a good compliment to the analyses. However, the extrapolation of the significance of this data to things such as the Daley Gap is not justified, and thus the significance of the paper is not as large as would appear from statements in the introduction and conclusion sections.

The manuscript focuses on the mechanism of immiscible liquids forming CBLs around crystals. This is now made clearer in the introduction and abstract and conclusion (see comments 2 and 26). Discussions linking this to the Daley gap have been largely removed from the manuscript. However, we retain a tentative statement in the discussion, eluding to the fact that our findings could help us understand the Daley gap (line 292); however, more work at different sample and resolution scales would be needed.

Šσ

Malcolm J Rutherford

- Findik F (2012) Improvements in spinodal alloys from past to present. *Materials & Design* 42:131-146 doi:<https://doi.org/10.1016/j.matdes.2012.05.039>
- Helz R (1980) Crystallization history of Kilauea Iki lava lake as seen in drill core recovered in 1967–1979. *Bulletin Volcanologique* 43:675-701
- Helz RT (1987) Differentiation behavior of Kilauea Iki lava lake, Kilauea Volcano, Hawaii: an overview of past and current work. *Magmatic processes: physicochemical principles* 1:241-258
- Helz RT, Clague DA, Sisson TW, Thornber CR (2014) Petrologic Insights into Basaltic Volcanism at Historically Active Hawaiian Volcanoes. In: Poland MP, Takahashi TJ, Landowski CM (eds) *Characteristics of Hawaiian volcanoes*. USGS Professional Paper 1801, pp 237-292
- Honour VC, Holness MB, Partridge JL, Charlier B (in review) Microstructural evolution of silicate immiscible liquids in ferrobasalts. *Contributions to Mineralogy and Petrology*
- Jean MM, Christiansen EH, Champion DE, Vetter SK, Phillips WM, Schuth S, Shervais JW (2018) Caldera Life-Cycles of the Yellowstone Hotspot Track: Death and Rebirth of the Heise Caldera. *Journal of Petrology* 59:1643-1670 doi:10.1093/petrology/egy074
- Neave DA, Buisman I, Maclennan J (2017) Continuous mush disaggregation during the long-lasting Laki fissure eruption, Iceland. *American Mineralogist* 102:2007-2021 doi:10.2138/am-2017-6015CCBY
- Neave DA, Passmore E, Maclennan J, Fitton G, Thordarson T (2013) Crystal–melt relationships and the record of deep mixing and crystallization in the ad 1783 Laki Eruption, Iceland. *Journal of Petrology* 54:1661-1690
- Philpotts A, Ague J (2009) *Principles of igneous and metamorphic petrology*. Cambridge University Press,

REVIEWERS' COMMENTS:

Reviewer #1 (Remarks to the Author):

The authors have done an impressive amount of work addressing the comments of all three reviewers. All of my specific comments have been more than adequately addressed. I think the paper is much clearer now, and I am even more convinced that this is an interesting and important phenomenon. It will be interesting to think about what samples can be analyzed in the future that could show signs of the CBLs actually migrating due to their density.

As the other two reviewers also note, I still think there is a question of whether these CBLs ever get large enough, at low enough crystallinities, to migrate. But I'm happy with how the paper discusses the issue.

I have one minor comment. In the mass/charge spectrum shown in the supplement, 28 is assigned to Fe²⁺. But it could also be Si¹⁺. Presumably the minor peaks look more like Fe than Si? Are the ratios exactly the Fe ratios? Or is there some contribution from Si? Any way to deconvolve the two? Likewise, 40 is assigned to Ca¹⁺, but could also be MgO¹⁺. How could these overlaps affect the Ca and Fe APT profiles? Maybe just a few sentences in the supplementary section on the m/q spectrum.

Reviewer #3 (Remarks to the Author):

The main purpose of this paper is to note and describe the occurrence of associated boundary layers adjacent to plagioclase and pyroxene phenocrysts and immiscible Fe- and Si-rich immiscible liquids in several basaltic lava flows including the well studied 1976 Kilauea Iki lava lake. The boundary layers next to these phenocrysts in samples quenched at 1050 to 1112 degrees C are similar in composition to the immiscible melts which typically form at lower temperatures. The paper argues convincingly that distribution and composition of these melts suggest the boundary layers facilitate the development of liquid immiscibility in these cooling lava flow tops at temperatures above those where experiments illustrate they can be produced by nucleation.

The revised paper has appropriately downplayed the extension of these conclusions to include the possibility this process occurs on a large scale to explain observed gaps (Daly) in the continuum of melt compositions in natural basaltic terrains. There just is not sufficient evidence that this process occurs early enough to facilitate large scale separation of these immiscible melts from each other given the crystal-rich character of most basaltic magmas where immiscibility is encountered, even though this paper suggests a possible mechanism to develop immiscibility at higher temperatures. The paper as now revised does contain an excellent description of the associated crystal boundary layer and immiscible melts in these samples, however; and makes a convincing case for the potential importance of the crystal boundary layer development in causing magma composition changes. It also contains an effective blend of theoretical analysis and review of existing experimental data to develop and constrain a model that explains the observation presented. I highly recommend this revised paper be accepted for publication.

Malcolm J Rutherford

REVIEWERS' COMMENTS:

REVIEWER #1 (REMARKS TO THE AUTHOR):

The authors have done an impressive amount of work addressing the comments of all three reviewers. All of my specific comments have been more than adequately addressed. I think the paper is much clearer now, and I am even more convinced that this is an interesting and important phenomenon. It will be interesting to think about what samples can be analyzed in the future that could show signs of the CBLs actually migrating due to their density.

We are pleased that Reviewer #1 thinks the paper is much clearer and is convinced by our results.

As the other two reviewers also note, I still think there is a question of whether these CBLs ever get large enough, at low enough crystallinities, to migrate. But I'm happy with how the paper discusses the issue.

We recognise this as a potential issue too and so discuss this in the paper.

I have one minor comment. In the mass/charge spectrum shown in the supplement, 28 is assigned to Fe²⁺. But it could also be Si¹⁺. Presumably the minor peaks look more like Fe than Si? Are the ratios exactly the Fe ratios? Or is there some contribution from Si? Any way to deconvolve the two? Likewise, 40 is assigned to Ca¹⁺, but could also be MgO¹⁺. How could these overlaps affect the Ca and Fe APT profiles? Maybe just a few sentences in the supplementary section on the m/q spectrum.

We have added two paragraphs to Supplementary Figure 17 discussing potential peak overlaps. We address each question from the reviewer – as detailed below.

In particular peak overlaps have been considered. The major isotopes of Fe are 56, 54, 57, 58, so the +2 charge states are at 28, 27, 28.5, and 29 in order. There are expected overlaps both with ²⁸Si and ²⁹Si as well as ²⁷Al. If one assumes some sort of natural abundance, then there are clear mixes of these elements/isotopes in those peaks. When reporting average composition for some volume (freely selected by the analyst) one can use these natural abundances to estimate the individual contributions (overlaps), but this is not possible on a single ion basis as the statistics of many ions are needed to correct an average composition for some sub-volume. For the figure below specifically, the intensity of the 29 peak suggests it is almost all silicon, since the ⁵⁸Fe abundance is relatively weak. Consequently, this suggests that ~20% of the ⁵⁶Fe⁺⁺ peak is actually silicon. Also, about half of the ²⁷Fe peak is Al and half is ⁵⁴Fe⁺⁺.

For Ca, almost all of the detected Ca is observed in the Ca⁺⁺ charge state. There is some overlap of the ⁴⁴Ca⁺⁺ with ²⁸Si¹⁶O⁺⁺ peak but it is dominated by SiO⁺⁺, so is negligible. For Ca⁺, there are interferences and upon closer scrutiny, the ⁴⁰Ca and ⁴²Ca peaks are almost certainly MgO⁺ at nearly 100%. These two peaks account for 0.06 at.% of the composition i.e. there is not Ca⁺. This change would reduce the estimated Ca by about 2% (relative not absolute; this very small change is probably less than the estimated error on the average composition measurement) and increase Mg by about 2%, making this mislabelling statistically insignificant.

REVIEWER #3 (REMARKS TO THE AUTHOR):

The main purpose of this paper is to note and describe the occurrence of associated boundary layers adjacent to plagioclase and pyroxene phenocrysts and immiscible Fe- and Si-rich immiscible liquids in several basaltic lava flows including the well studied 1976 Kilauea Iki lava lake. The boundary layers next to these phenocrysts in samples quenched at 1050 to 1112 degrees C are similar in composition to the immiscible melts which typically form at lower temperatures. The paper argues convincingly that distribution and composition of these melts suggest the boundary layers facilitate the development of liquid immiscibility in these cooling lava flow tops at temperatures above those where experiments illustrate they can be produced by nucleation.

We are pleased that Reviewer #3 is convinced by our results.

The revised paper has appropriately downplayed the extension of these conclusions to include the possibility this process occurs on a large scale to explain observed gaps (Daly) in the continuum of melt compositions in natural basalt terrains. There just is not sufficient evidence that this process occurs early enough to facilitate large scale separation of these immiscible melts from each other given the crystal-rich character of most basaltic magmas where immiscibility is encountered, even though this paper suggests a possible mechanism to develop immiscibility at higher temperatures.

We recognise this as a potential issue too and so discuss this in the paper.

The paper as now revised does contain an excellent description of the associated crystal boundary layer and immiscible melts in these samples, however; and makes a convincing case for the potential importance of the crystal boundary layer development in causing magma composition changes. It also contains an effective blend of theoretical analysis and review of existing experimental data to develop and constrain a model that explains the observation presented. I highly recommend this revised paper be accepted for publication.

We are delighted that Reviewer #3 thinks our paper has an effective blend of theory and existing data to constrain our new model.

Malcolm J Rutherford